

# Regional-scale brine migration along vertical pathways due to $CO_2$ injection – Part 2: a simulated case study in the North German Basin

Alexander Kissinger[1], Vera Noack[2], Stefan Knopf[2], Wilfried Konrad[3], Dirk Scheer[3], and Holger Class[1]

[1]Department of Hydromechanics and Modelling of Hydrosystems, University Stuttgart, Pfaffenwaldring 61, 70569 Stuttgart, Germany
[2]Bundesanstalt für Geowissenschaften und Rohstoffe (BGR), Stilleweg 2, 30655 Hannover
[3]DIALOGIK, Lerchenstraße 22, 70176 Stuttgart, Germany

*Correspondence to:* Alexander Kissinger (alexander.kissinger@iws.uni-stuttgart.de)

**Abstract.** Saltwater intrusion into potential drinking water aquifers due to the injection of $CO_2$ into deep saline aquifers is one of the hazards associated with the geological storage of $CO_2$. Thus, in a site-specific risk assessment, models for predicting the fate of the displaced brine are required. Practical simulation of brine displacement involves decisions regarding the complexity of the model. The choice of an appropriate level of model complexity depends on multiple criteria: the target variable of interest, the relevant physical processes, the computational demand, the availability of data, and their uncertainty. In this study, we set up a regional-scale geological model for a realistic (but not real) on-shore site in the North German Basin with characteristic geological features for that region. A major aim of this work is to identify the relevant parameters controlling saltwater intrusion in a complex structural setting and to test the applicability of different model simplifications. The model that is used to identify relevant parameters fully couples flow in shallow freshwater aquifers and deep saline aquifers. This model also includes variable-density transport of salt and realistically incorporates surface boundary conditions with groundwater recharge. The complexity of this model is then reduced in several steps, by neglecting physical processes (two-phase flow near the injection well, variable-density flow) and by simplifying the complex geometry of the geological model. The results indicate that the initial salt distribution prior to the injection of $CO_2$ is one of the key parameters controlling shallow aquifer salinization. However, determining the initial salt distribution involves large uncertainties in the regional-scale hydrogeological parametrization and requires complex and computationally demanding models (regional-scale variable-density salt transport). In order to evaluate strategies for minimizing leakage into shallow aquifers, other target variables can be considered, such as the volumetric leakage rate into shallow aquifers or the pressure buildup in the injection horizon. Our results show that simplified models, which neglect variable-density salt transport, can reach an acceptable agreement with more complex models.

## 1 Introduction

Any effort in investigating and developing the Carbon Dioxide Capture and Storage technology (CCS) unavoidably touches the social and political sphere and needs to take into account the broader societal debate. From the very beginning, this research on brine migration, was aimed at involving expert and stakeholder knowledge in the simulation of impacts during the injection of $CO_2$ into deep saline aquifers. Therefore, this work is split into two papers (Part 1 and Part 2), where Part 1 deals with





the concept of "Participatory Modeling" as a means to involve external experts and stakeholders in the modeling process, and Part 2 deals with the technical findings relevant for modeling brine migration. The Participatory Modeling process influenced the setup of the geological model and the scenarios presented in this paper.

Successful geological storage of $CO_2$ on a climate-relevant scale has been shown, for example at the Sleipner site (Skalmer-aas, 2014), and there are currently 15 large-scale CCS projects in operation (Global CCS Institute, 2016). Further large-scale projects are needed to meet the estimated storage demand in the near future (Bruckner et al., 2014; Eom et al., 2015) and to improve the understanding of a safe and efficient storage (IEA, 2013). Identifying possible storage sites generally involves a multi-stage process, where different criteria are evaluated, like storage safety, storage efficiency, and economical feasibility.

The injection of super-critical $CO_2$ into saline aquifers inevitably leads to the displacement of resident brine. Hazardous situations may arise if brine migrates vertically through discontinuities like permeable fault zones or improperly plugged abandoned wells into shallow aquifer systems, where the brine may contaminate drinking water. Salt concentrations at a drinking-water production well should not rise above the regulatory limits and eventually lead to a shutdown of production.

The extent of pressure propagation was already subject of several simulation studies. Models have shown that the area where brine migration can occur is much larger than the actual extent of a $CO_2$-plume (2-8 km), as elevated pressures are predicted up to 100 km from the injection well within the injection horizon (Birkholzer et al., 2009; Birkholzer and Zhou, 2009; Schäfer et al., 2011). Schäfer et al. (2011) performed simulations in a geological system consisting of aquifer and barrier formations bounded by a sealing fault zone. Birkholzer et al. (2009) considered a multi-layered system consisting of a sequence of horizontal aquifers and aquitards and investigated both lateral and vertical pressure propagation. They conclude that leakage across aquitards should be considered for realistic pressure propagation. However, they do not expect significant damage due to vertical brine migration unless vertical pathways such as permeable fault zones or improperly plugged abandoned wells exist where focused leakage may occur. More recent studies focus on the simplification of the simulation tools for quantifying brine migration and developing pressure management tools. Brine leakage through improperly plugged abandoned wells was investigated in Celia et al. (2011) using a semi-analytical model described in Celia and Nordbotten (2009) and Nordbotten et al. (2009). A comparison of models of varying complexity on the basin scale with multiple injection wells was conducted by Huang et al. (2014). They concluded that single-phase numerical models are sufficient for predicting basin-scale pressure response. Analytical and semi-analytical solutions depending on superposition of solutions in time and space may not be accurate enough as the variability of formation properties (heterogeneity and anisotropy) cannot be captured. Cihan et al. (2011) developed an analytical model capable of handling multi-layered systems considering diffuse leakage (through aquitards) and focused leakage (abandoned well and fault zones). The same analytical model is also applied in Birkholzer et al. (2012), where pressure-management strategies are compared. Zeidouni (2012) presented an analytical model for determining brine flow through a permeable fault zone into aquifers separated by impermeable aquitards. This model has a realistic description of the fault zone, as lateral and vertical transmissivity within the fault zone can be assigned independently of each other, thereby allowing a wide range of fault zone configurations. Oldenburg and Rinaldi (2010) set up an idealized numerical model of two aquifers separated by a barrier layer and connected by a permeable fault zone. Their results show that a new hydrostatic equilibrium may establish if saltwater is pushed upwards through the fault zone due to an increase in pressure in the lower





aquifer. The new equilibrium depends on the salt concentration in the lower aquifer, where low concentrations may cause continuous upward flow as opposed to high salt concentrations. Tillner et al. (2013) consider brine-migration scenarios for a potential storage site in northern Germany using a multi-phase (brine and supercritical $CO_2$) multi-component ($H_2O$, NaCl, and $CO_2$) model accounting for salt-dependent density differences. They included several permeable and impermeable fault

zones, thereby controlling leakage into overlying aquifers. They conclude that the choice of boundary conditions for the lateral boundary has the highest impact on the observed brine migration, while the results are less sensitive to the fault permeability. The model for the deep subsurface used by Tillner et al. (2013) was coupled (one-way coupling) to a model comprising shallow freshwater aquifers (Kempka et al., 2013) using flow through the fault zones as boundary conditions for the shallow aquifer model. The results indicate that an increase in the salt concentration due to $CO_2$-injection is only recognizable in areas with an

already elevated, natural salt concentration. Walter et al. (2012, 2013) use a generic horizontally stratified multi-layer system with a circular fault zone surrounding the injection well at a certain distance. They also use a compositional multi-phase model (water, supercritical $CO_2$, NaCl) to calculate the brine flow into a shallow aquifer. Walter et al. (2012) assume a constant initial salt concentration across the deep layers, while in Walter et al. (2013), they assume a linear increase of the salt concentration with depth. The results show that the amount of salt entering the shallow aquifer varies significantly between these two as-

sumptions, with much more salt entering in the constant concentration case. Therefore, the calculation of salt transport into shallow aquifers is not only uncertain with respect to the boundary conditions and hydrogeological parametrization but also with respect to the initial salt concentration in the system. Tillner et al. (2016) confirm this finding, while additionally stressing the importance of the fault-damage-zone volume determining the intensity of the salinization of shallow freshwater aquifers.

For this study we used data from a 3D geological structure in the North German Basin. This basin has been previously iden-

tified as the most relevant region regarding $CO_2$ storage capacity in Germany (Knopf et al., 2010). The geological model comprises layers from the injection horizon to shallow freshwater aquifers. In contrast to earlier work, for example by Kempka et al. (2013), our model fully couples flow in shallow freshwater aquifers with deep saline aquifers. The research questions we would like to address with this model are the following: which parts of the shallow aquifers are prone to salinity increases, and which are the relevant parameters controlling saltwater migration into shallow aquifers in such a complex structural setting?

We then analyze the effects of reducing model complexity by neglecting physical processes such as two-phase flow near the injection well or variable-density flow. The model of lowest complexity in this study is the analytical solution by Zeidouni (2012). There are two primary reasons for reducing model complexity. First, reduced computational costs allow more realizations of the model to analyze the inherent uncertainty of the hydrogeological data. Second, less complex models generally need less data, which is good in cases where many data are uncertain. The research question we would like to address with this

model comparison is: how far can we reduce the complexity before the models become too simple?

Section 2 introduces the North German Basin and the regional-scale geological model used in our investigations. Section 3 gives a brief overview of the different models used as well as an explanation of the boundary and initial conditions applied. Sec. 4 presents the results; first, different target variables with respect to brine migration into shallow aquifers are defined, followed by simulation results, where important parameters of the system are varied. This is followed by the analysis on the

reduction of model complexity. Finally, the results are discussed in Sections 5 and 6.





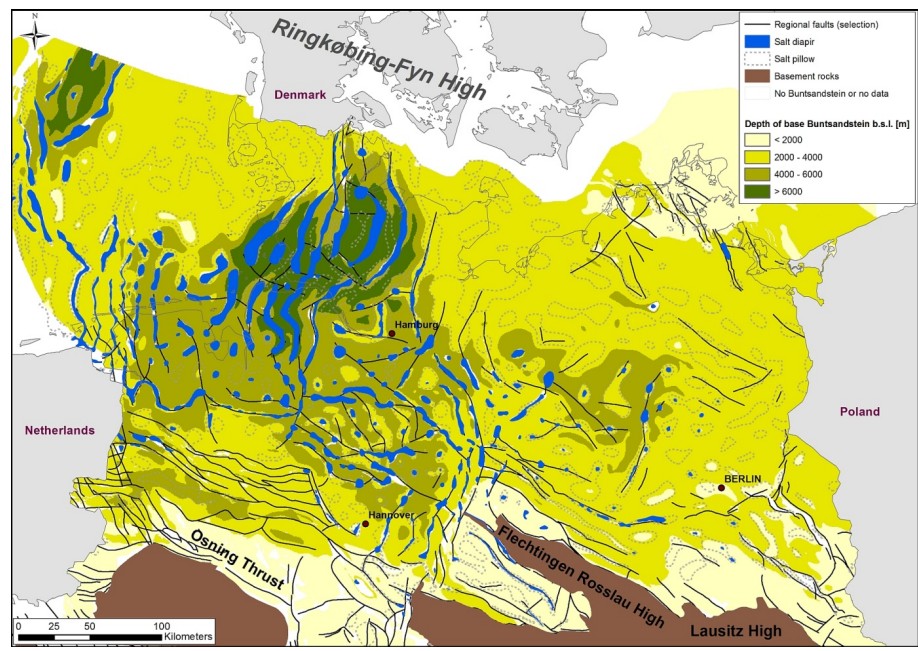

**Figure 1.** Extent and depth of the Buntsandstein group and important structural elements in the North German Basin (depicting data only for the German mainland and the German sectors of the North Sea and the Baltic Sea). Based on Reinhold et al. (2008), Doornenbal and Stevenson (2010), Schulz et al. (2013).

## 2 Geological model

### 2.1 Geology of the North German Basin

The North German Basin (NGB) is part of the Central European Basin System (CEBS), a continental rift system which extends from the North Sea, across the Netherlands, Northern Germany, Denmark, and towards Poland (Mazur and Scheck-Wenderoth,

2005). The NGB is one of the main Permian sub-basins of the CEBS and represents the southern margin of this basin system (Mazur and Scheck-Wenderoth, 2005; Cacace et al., 2008). Sediments of up to 12 km thickness have been locally accumulated within the NGB (Kockel, 1998), representing deposits from the Permian to the Cenozoic age of various lithologies.

The deposits within the NGB contain regionally important reservoir rocks (sandstones) and barrier rocks (shale, evaporates), which are prerequisites for the safe storage of buoyant fluids. Potential reservoir and barrier rock units of the NGB have been

evaluated in recent years by several projects (e.g. Reinhold et al. (2011); Jähne-Klingberg et al. (2014)). Accordingly, potentially suitable Permian Upper Rotliegend- and Triassic Middle Buntsandstein reservoir rock units are widely spread across the NGB, thus holding the bulk of subsurface storage potential (see Fig. 1). In contrast, the areal distribution of stratigraphic younger reservoir rock units from the Upper Triassic, Middle Jurassic, and Lower Cretaceous are considerably restricted (Reinhold et al., 2011). Next to various potentially suitable barrier rocks from Permian and Mesozoic strata, a special focus may be

given to the Oligocene Rupelian clay, which forms an important regional hydraulic barrier between shallow freshwater aquifers





and deep saline aquifers in the NGB (Reinhold et al., 2011).

The sedimentary cover of the NGB has been influenced by salt tectonics since the Triassic (Maystrenko et al., 2008). Mobilization of Zechstein salt affected the sedimentation and deformation of the Mesozoic and Cenozoic strata within the basin. Salt tectonics led to the development of ca. 450 salt structures in the NGB (Reinhold et al., 2008), which have either bent upward

(salt pillows) or penetrated the overburden (salt diapirs and salt walls). In general, the complex structural evolution of the NGB resulted in primarily NW- and N-trending structures (Kley et al., 2008). These trends can be followed in faults, folds and salt diapirs, or along salt walls on a basin and sub-basin scale. The evolution of the NGB favored the formation of a multitude of geological structures within Mesozoic strata that may act as traps (e.g. anticlinal traps and fault traps) for the storage of buoyant fluids. Many natural oil and gas fields are indicators of these favorable reservoir conditions, as they proof the presence

of suitable reservoir and barrier-rock units, as well as their suitable trap structures.

## 2.2   Regional-scale geological model

The geological model described here is not a real site, but based on a real structural configuration derived from the German North Sea and combined with groundwater isoline data from a shallow freshwater aquifer in the federal state of Brandenburg. The model comprises layers from the deep saline injection horizon up to shallow freshwater aquifers.

Geological data from 3D models of a southwestern German North Sea region are used as a database for the structural model of the deep subsurface (Bombien et al., 2012; Asprion et al., 2013; Kaufmann et al., 2014; Wolf et al., 2014). The region belongs to the NGB and was affected by salt mobilization during different geological time periods. In this area, salt mobilization led to the rise of a salt diapir and the formation of anticlinal structures. Thereby, the typical geological units of Muschelkalk, Keuper, and Jurassic have not been deposited. However, the lithological composition of the accumulated geological units, and

their structural configurations, represent excellent conditions for structural traps. The database from the southwestern German North Sea provides depth lines of stratigraphical surfaces that are used to construct main geological units of the NGB in the 3D structural model. Hence, the stratigraphic succession of the NGB is represented in a simplified fashion in this study. The following eight sets of depth lines of stratigraphical surfaces are available to construct the layers for the 3D structural model: base and top of Zechstein (Permian), base of Middle Buntsandstein (Triassic), base of Upper Buntsandstein (Triassic), top of

the Buntsandstein (Triassic), base of Upper Paleocene (Tertiary), base of Oligocene (Tertiary) and base of Quaternary. From these datasets, 2D grid surfaces for the respective geological units of the model layers are interpolated using the convergent interpolation technique (software Petrel 2012.1). The resulting geological model and its dimensions are shown in Fig. 2.

The depth of the base of the Zechstein varies only slightly across the model domain, ranging from depths of 3300 m to 4000 m. In contrast, the depth of the top of the Zechstein shows a highly differentiated structural pattern due to the mobilization of the

Zechstein salt, and varies in depths between 3800 m and 350 m. This mobilization also affected the geometry of the overburden. The result is an elongated anticlinal structure of Mesozoic sediments on top of a salt pillow (Permian Zechstein salt). This dome structure descends gently into a structural low (Syncline). The latter is bordered by an elongated steeply rising salt wall (diapir), as shown in Fig. 3. In order to reflect geological and hydrogeological conditions of a storage complex consisting of a storage horizon and rock barrier systems, we add virtual surfaces for important geological units to the final model. The Meso-




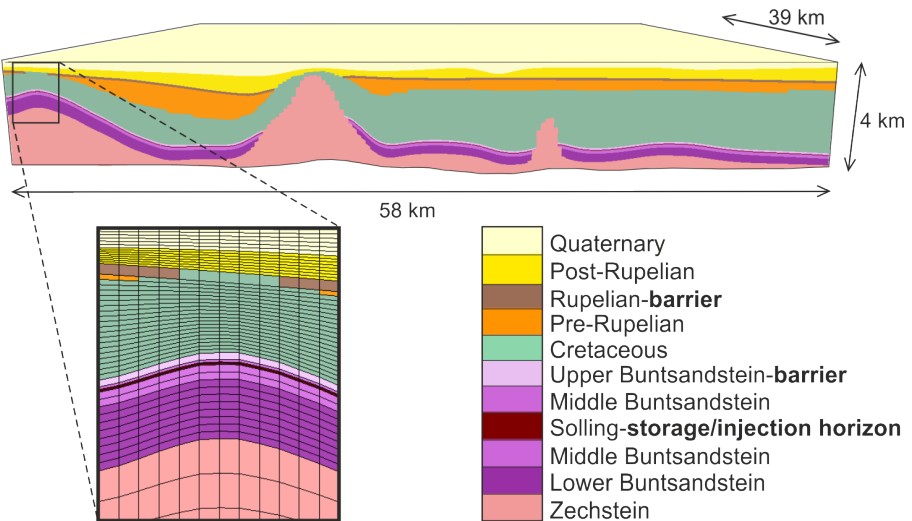

**Figure 2.** Perspective view on the 3D geological model with zoom in on the anticlinal structure showing the mesh of the 3D volume model. Vertical exaggeration 2:1.

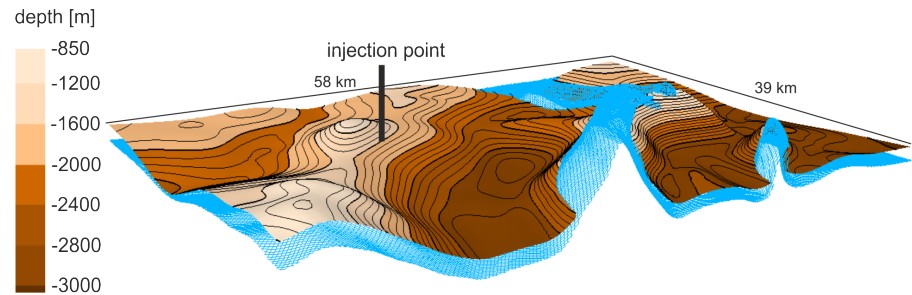

**Figure 3.** Depth contour map of the top of the Solling injection horizon. The top of the Zechstein salt is displayed with the blue mesh, with the salt wall piercing through the injection horizon. The injection point at the flank of the anticlinal structure in about 1600 m depth is projected on top of the Solling storage horizon. Vertical exaggeration is 2:1.



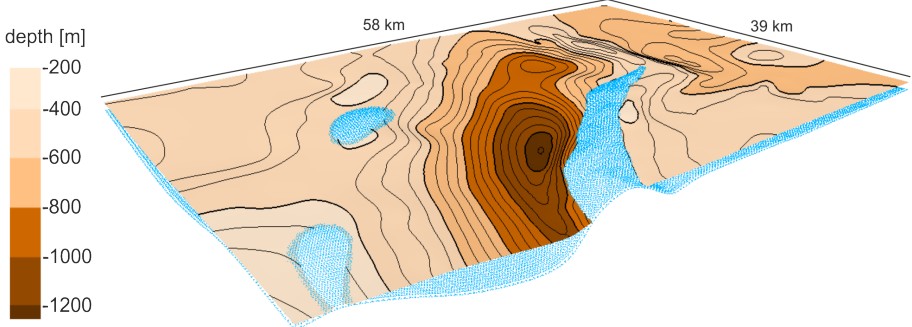

**Figure 4.** Depth contour map of the top of the Rupelian clay barrier with discontinuities where Cretaceous sediments penetrate the Rupelian clay barrier. The top of the Cretaceous is displayed as blue mesh. Vertical exaggeration is 2:1.

zoic sediments above the mobilized Permian Zechstein salt include the storage horizon and the barrier rocks. Modifications of the model affect the layers for the Middle Buntsandstein, where we added two surfaces, representing top and bottom of the Solling Sandstone, a unit that is considered in this study as the injection horizon for $CO_2$. The injection point is situated at the flank of the anticlinal structure, as indicated in Fig. 3, which means that once injected, the supercritical $CO_2$ will migrate

upwards along the anticlinal structure and finally accumulate beneath the dome structure. The geological layer overlying the rock unit of the Middle Buntsandstein represents the rock unit of the Upper Buntsandstein. The Upper Buntsandstein is the first important barrier in the system, preventing fluid migration out of the storage horizon. It will therefore be referred to here as Upper Buntsandstein barrier. The Cenozoic sediments include the Rupelian clay barrier and the freshwater complex. The base of Oligocene is deemed to be the base of the hydraulically important Rupelian clay barrier, which is the second barrier in

the geological model, separating shallow freshwater aquifers from deep saline aquifers. We modified this hydraulic barrier to be penetrated by the uplifted Cretaceous sediments on top of the anticlinal structure. Such discontinuities (so-called hydrogeological windows) are also present on top of the rising salt wall where the diapir and overlying Cretaceous sediments pierce into the Rupelian clay barrier (see Fig. 4). Also the Tertiary post-Rupelian and the Quaternary are pierced by the lifted sediments. For the top of the geological model, we used a dataset containing the groundwater isolines of an upper freshwater aquifer in

the state of Brandenburg (data provided by LUGV (2012)).

Data for lithological composition and the corresponding parameters are derived from regional literature data and simulation studies (Larue (2010); Reutter (2011); Schäfer et al. (2011); Noack et al. (2013)). Table 1 shows the main lithological compositions, the average thicknesses of the layers, the porosity, and the permeability data assigned to the model layers. Each layer is assumed to be homogeneous in permeability and porosity. Since the Zechstein layer can be considered impermeable it is not

considered in the simulations, except for the salt wall. To establish a more realistic base flow in the shallow aquifers, we split the Quaternary layer into two parts, where the uppermost layer Quaternary 1 has the highest permeability of all layers (see Table 1).

Making a conservative assumption, we assume a permeable vertical pathway along the whole flank of the salt wall and refer to



**Table 1.** Properties of the model layers according to Larue (2010); Reutter (2011); Schäfer et al. (2011); Noack et al. (2013).

| Layer | Lithology | Thickness [m] | Porosity [%] | Permeability [m$^2$] |
|---|---|---|---|---|
| Quaternary 1 | sand, gravel | 100 | 20 | $6 \cdot 10^{-11}$ |
| Quaternary 2 | sand, gravel | 200 | 20 | $1 \cdot 10^{-12}$ |
| (Tertiary) Post-Rupelian | sand, silt | 400 | 15 | $1 \cdot 10^{-13}$ |
| (Tertiary) Rupelian clay barrier | clay | 80 | 10 | $1 \cdot 10^{-18}$ |
| (Tertiary) Pre-Rupelian | sand, sandstone | 350 | 10 | $1 \cdot 10^{-13}$ |
| Cretaceous | chalk, claystone | 900 | 7 | $1 \cdot 10^{-14}$ |
| Upper Buntsandstein barrier | salt, anhydrite, claystone | 50 | 4 | $1 \cdot 10^{-18}$ |
| Middle Buntsandstein | siltstone | 20 | 4 | $1 \cdot 10^{-16}$ |
| Solling | sandstone | 20 | 20 | $1.1 \cdot 10^{-13}$ |
| Middle Buntsandstein | siltstone | 110 | 4 | $1 \cdot 10^{-16}$ |
| Lower Buntsandstein | clay- and siltstone | 350 | 4 | $1 \cdot 10^{-16}$ |
| Permian Zechstein | rock salt | - | 0.1 | $1 \cdot 10^{-20}$ |
| Fault zone | - | 50 | 30 | $1 \cdot 10^{-12}$ |

it as a fault zone. This fault zone is a permeable connection between the injection horizon and the shallow aquifers above the Rupelian clay barrier. The assumption of fluid migration via vertical pathways in sediments flanking salt structures is a matter of debate. After LBEG (2012) "the contact zone between salt domes and the CO2 -sequestration horizon is assumed to be a zone of weakness, similar to geological faults". Such zones of weakness may provide effective vertical migration pathways.

To our understanding, the assumption of a permeable fault zone along the whole flank of a diapir is an exaggeration of real geological conditions. In contrast, faults of smaller range at shallower depths in the sediments on top of the hanging wall of diapirs may provide pathways for fluids. Table 1 shows the fault zone permeability and porosity of the reference setting. Subsequently, the twelve modified 2D grids confining the eleven geological layers of the geological model were merged into a consistent 3D structural model. In the structural gridding process we assigned a consistent cell size of 300 x 300 m horizon-

tally to the 3D hexahedron mesh (Fig. 2). The vertical resolution depends on the thickness of each layer resolved in the model. In order to sufficiently reproduce the complex geometry we subdivided all layers of large thicknesses resulting in a vertical resolution that varies between 10 and 160 m.

## 3  Numerical and analytical models

All models with different conceptual complexity regarding the implemented physics and the geometry of the geological model,

except the analytical solution by Zeidouni (2012), are implemented in the open-source numerical simulator DuMu$^x$ (Flemisch



et al., 2011; Schwenck et al., 2015). DuMu$^x$ was already used in previous CCS-related publications (Darcis et al., 2011; Walter et al., 2012, 2013; Kissinger et al., 2014) and code comparison studies (Nordbotten et al., 2012; Class et al., 2009).

## 3.1 Model types

For the analysis of model simplifications, four different models will be used to investigate brine migration in the geological setting (Sec. 2.2). The single-phase (brine), two-component (water and salt) model, referred to here as 1p2c model, serves as the reference model which accounts for variable-density salt transport. The model simplifies the injection process, where a volume-equivalent rate of brine is injected instead of $CO_2$, thereby neglecting compressibility effects and movement of the supercritical $CO_2$ near the injection well. The single-phase (brine) single-component (water) model, referred to here as 1p1c model, is a simplification of the 1p2c model, where salt transport is neglected; salt is instead considered to be a pseudo component in terms of locally varying salinities (constant in time) within the domain affecting the fluid properties (density and viscosity). This is similar to the constant geothermal gradient which we also apply. The third model accounts for two-phase flow (brine and $CO_2$) as well as three-component (water, $CO_2$ and salt) transport, referred to as the 2p3c model. This model is the most complex model considered here, as it takes into account the injection and transport of $CO_2$ near the injection point as well as the variable-density salt transport. The 1p1c, 1p2c, and the 2p3c models are implemented in the numerical simulator DuMu$^x$. The balance equations along with more detailed explanations are given in the Appendix A. The fourth model used in this work is the analytical solution presented in Zeidouni (2012). It will be referred to as the "Analytical Model". It accounts for single-phase single-component flow in a horizontally stratified system of aquifers, separated by completely impermeable barrier layers. The aquifers are coupled through a permeable fault zone. The Analytical Model cannot account for diffuse leakage over the barrier layers. Further information regarding the setup and the boundary conditions of the Analytical Model are given in Appendix B. An overview of the different models and the processes they neglect is given in Table 2.

**Table 2.** Overview of the different models and the processes which are neglected.

|  | 2p3c | 1p2c | 1p1c | Analytical |
| --- | --- | --- | --- | --- |
| Two-phase flow around injection | X | – | – | – |
| Variable-density salt transport | X | X | – | – |
| Complex geometry | X | X | X | – |
| Diffuse leakage across barrier layers | X | X | X | – |





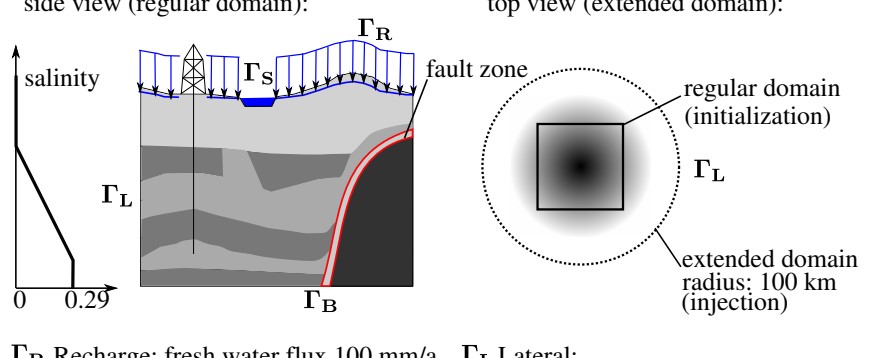

$\Gamma_R$ Recharge: fresh water flux 100 mm/a     $\Gamma_L$ Lateral:
$\Gamma_S$  Stream: Atm. pressure, zero salinity          initialization run - no flow (regular)
$\Gamma_B$ Bottom: no flow for water, sal. = 0.29       injection run - infinite aquifer (extended)

**Figure 5.** Boundary and initial conditions of the domain shown through a simplified sketch of the geological model. A linear salinity profile increasing with depth up to a maximum salinity (salt mass fraction) of $0.29\,\mathrm{kg\,kg^{-1}}$ is assumed as an initial condition for the initialization run. Also shown is the position of the fault zone situated at the flank of the salt wall in red.

### 3.2 Initial and boundary conditions

Realistic boundary and initial conditions are required for modeling regional-scale brine displacement. The boundary conditions for the numerical model are shown in Fig. 5. On the top boundary ($\Gamma_R$), a constant recharge of $100\,\mathrm{mm\,year^{-1}}$ is set (Neumann boundary condition) except for the nodes close to a river ($\Gamma_S$), where a constant atmospheric pressure is set (Dirichlet boundary

condition). In order to obtain a realistic base flow in the shallow freshwater aquifers, we use the data of the main rivers from the catchment area associated with the groundwater isolines dataset, which form the top of the model domain and are discussed above (Sec. 2.2). The rivers act as a sink in the system. Figure 6 shows the top view of the domain, with the location of the rivers and the elevation of the groundwater isolines. Note that the differences in the groundwater table are rather small ($17\,\mathrm{m}$). It is assumed that full hydraulic contact exists between the rivers and the groundwater. We performed a stationary calibration

to match the simulated pressure distribution on the top of the domain with the groundwater isolines, which resulted in the increased permeability of the uppermost layer, Quaternary 1 (see Table 1). A constant geothermal gradient of $0.03^\circ\mathrm{C\,m^{-1}}$ is assumed, starting from $8^\circ\mathrm{C}$ at the top of the domain.

In order to obtain a quasi-stationary salinity distribution prior to the $CO_2$ injection, an initialization run is required. The initial salt distribution below the Rupelian Clay Barrier is assumed to follow a linear increase of salinity with depth, with a maximum

salinity of $0.29\,\mathrm{kg\,kg^{-1}}$, see Fig. 5 (left). For the 1p1c model, with the salinity as a pseudo component, a steady state can be established within a single time step. However, for the models considering variable-density salt transport (1p2c and 2p3c) the initialization run is carried out for a period of 300,000 years, after which a system state has established that can be considered quasi-stationary on the time scale of the injection and post-injection, i.e. 100 years. During the initialization run, the bottom and lateral boundaries are closed. However, salt may enter the system through the bottom boundary or along the salt wall,

where a fixed maximum salinity of $0.29\,\mathrm{kg\,kg^{-1}}$ is set.





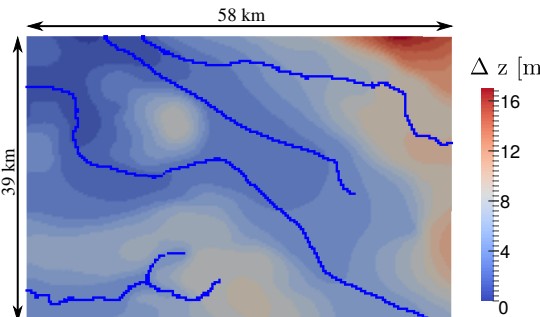

**Figure 6.** Top view on the groundwater table. The rivers are highlighted in blue. The elevation values are normalized to the minimum elevation of the groundwater table (data for groundwater isolines provided by LUGV (2012), data for rivers provided by LUGV (2014)).

The results from the initialization serve as initial conditions for the injection runs. During the injection runs, the domain is extended laterally for layers with a permeability greater than $1 \cdot 10^{-15}$ m$^2$ beneath the Rupelian barrier (i.e. the layers Pre-Rupelian, Cretaceous and Solling). The layers are extended to a distance of $100$ km from the center of gravity of the regular domain. This distance is sufficient for the boundaries of the regular domain to act as quasi-infinite aquifers.

## 4 Results

This section is sub-divided into three parts. (i) The target variables used in the simulations are briefly discussed, (ii) the results of the scenario analysis are presented, and (iii) the models of varying complexity are compared.

### 4.1 Definition of target variables

In order to compare the results, different target variables are used:

- **Flow into shallow aquifers:** Everything above the Rupelian clay barrier is defined here as shallow aquifers. Different areas over which the flow of salt, brine or water volumes is summed up are distinguished: (i) flow near the salt wall, comprising flow through the fault zone and flow through the Cretaceous dragged up by the salt wall (for ease of notation, we refer to both as flow through the fault zone); (ii) flow through the hydrogeological windows in the Rupelian clay barrier; and (iii) total flow into the shallow aquifers comprising (i) and (ii) as well as the flow through the intact Rupelian clay barrier. Figure 7a shows a view on the interface between the Rupelian clay barrier and the shallow aquifers. Further, the total salt flow into the more shallower Quarternary 2 and Quarternary 1 is also considered.

- **Pressure buildup at selected locations:** The simulated pressure buildup due to the injection is observed at two measurement points (M1 and M2) in the injection horizon Solling. The two points are on a straight line between the injection point and the nearest point on the salt wall: (i) M1 approximately 6 km from the injection and (ii) M2 approximately 13.5 km from the injection directly in the fault zone (see Fig. 7 b).





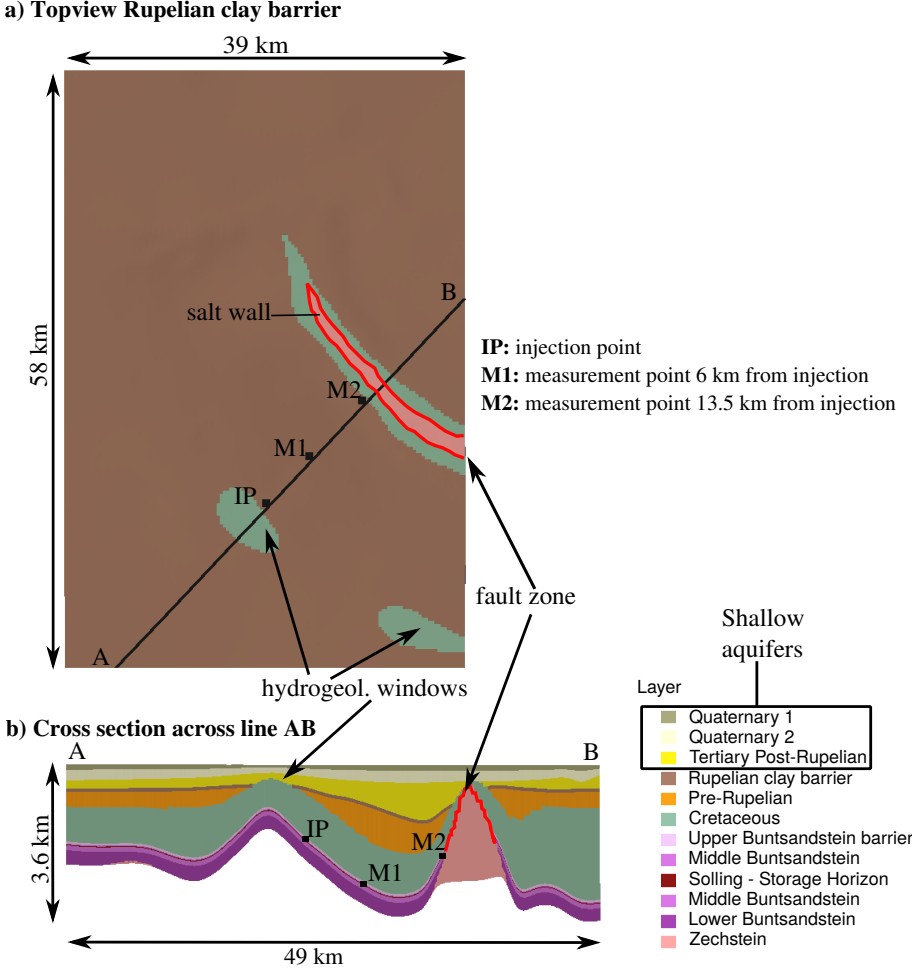

**Figure 7. a)** Top view on the Rupelian clay barrier. Also shown are the two hydrogeological windows in the Rupelian clay barrier layer and the salt wall piercing through the barrier layer. The fault zone is highlighted in red. **b)** shows a cross section (vertical exaggeration 4:1) across line A-B with approximate locations of the injection point (IP) and the two measurement points for pressure (M1 and M2).

    – **Concentration changes in shallow aquifers:** The injection-induced changes in the salt concentration will be shown on the top of the Rupelian clay barrier (as shown in Fig. 7 a), and on the top of the Post-Rupelian.

## 4.2 Identification of relevant parameters

In the following, four different scenario studies are evaluated, each varying a key parameter (initial salt distribution prior to the injection, lateral boundary conditions, role of the Upper Buntsandstein barrier permeability, and the fault zone transmissivity). All scenarios are evaluated against a reference model. The reference model is not understood as the most likely geological setup, but simply shows all processes under investigation on a recognizable scale. Porosities and permeabilities of the refer-





**Table 3.** List of parameters for the reference setting. The two phase flow specific parameters are only required for the 2p3c model.

| Parameter | Unit | Value |
|---|---|---|
| Compressibility solid phase | $\mathrm{Pa}^{-1}$ | $4.5 \cdot 10^{-10}$ |
| Depth injection | m | 1651 |
| Temperature gradient | $\mathrm{Km}^{-1}$ | 0.03 |
| Temperature top | K | 281.15 |
| Density $CO_2$ at injection point | $\mathrm{kgm}^{-3}$ | 686.5 |
| Density brine at injection point | $\mathrm{kgm}^{-3}$ | 1078 |
| Injection rate $CO_2$ | $\mathrm{kgs}^{-1}$ | 15.86 (0.5 Mt year$^{-1}$) |
| Volume-equivalent injection rate brine | $\mathrm{kgs}^{-1}$ | 24.95 |
| Recharge at top boundary | $\mathrm{mm\,year}^{-1}$ | 100 |
| Initial salinity gradient | $\mathrm{gL}^{-1}(100\mathrm{m})^{-1}$ | 15 |
| Maximum salinity | $\mathrm{kg}^{\mathrm{NaCl}}(\mathrm{kg}^{\mathrm{Brine}})^{-1}$ | 0.29 |
| Two-phase flow specific parameters (2p3c model ): | | |
| Brooks and Corey shape parameter $\lambda$ | - | 2.0 |
| Residual water saturation | - | 0.2 |
| Residual $CO_2$ saturation | - | 0.05 |

ence model are given in Table 1. The other relevant parameters for the reference model are given in Table 3. All simulations are performed with the 1p2c model, where brine is injected instead of $CO_2$ (see Table 2) at a constant rate. The effect of neglecting two-phase flow near the injection is discussed in the next section, where models of varying complexity are compared.

## 5    Scenario study 1: Initial salt distribution

We investigate here the effect of different salt distributions within the model domain prior to the injection. A linear increase of salinity over depth serves as the initial condition for the initialization run. It starts at 645 m, which is the average depth of the Rupelian clay barrier layer (i.e. the layer separating freshwater from saltwater). Once the maximum salinity is reached at a certain depth, the salinity does not increase further.

Three different scenarios with different salinity gradients are considered: Low, Medium, and High. The gradient is decreased to 10 (Low) and increased to 20 $\mathrm{gL}^{-1}(100\mathrm{m})^{-1}$ (High) from the reference value of 15 $\mathrm{gL}^{-1}(100\mathrm{m})^{-1}$ (Medium). First, we look at the state of the system after the initialization run for the Medium case in Fig. 8. Here, a quasi-stationary salt distribution has established. The salt distribution has considerably changed in the shallow aquifers from the initial salt gradient. The less dense brine has migrated above the Rupelian clay barrier due to the base flow which is controlled by the recharge boundary conditions and the position of the rivers. The initialization run shows that upconing occurs in the shallow aquifers near rivers. The rivers represent sinks because the lowest potential in the system (atmospheric pressure, zero salinity) is assigned there. The





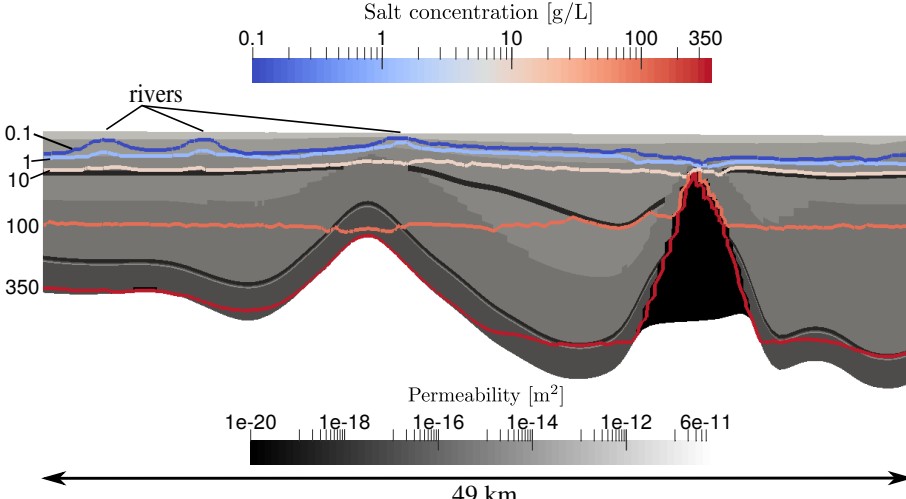

**Figure 8.** Salt distribution for the medium (reference) case after 300,000 years initialization run along the cross-section AB shown in Fig. 7 b) (vertical exaggeration 4:1). Six concentration-isolines are shown which correspond to the entries in the legend (0.1, 1, 10, 100 and 300). The permeability of the different layers is also shown. Please note the logarithmic scale of concentration and permeability.

10 g/L isoline closely follows the Rupelian clay barrier layer, with one exception being the region near the salt wall where the depth of the Rupelian clay barrier layer strongly increases. In the initialization run, the largest changes in the salt distribution are observed during the first 50,000 years.

The concentration increase after 50 years of injection (i.e. the end of the injection) is shown in Fig. 9. A concentration increase

of up to 3 g/L occurs at the top of the Rupelian clay barrier. The maximum concentration increase on the top of the Tertiary Post-Rupelian is an order of magnitude smaller (0.25 g/L). The largest changes occur close to the fault zone and at the hydrogeological window above the injection horizon. The change in concentration related to the injection increases from the Low to the High scenario, as the salt concentrations near the Rupelian clay barrier prior to the injection are higher. As a result, more salt can be displaced by the injection. This is illustrated in more detail in Fig. 10. Here, the total cumulative salt flow into

each of the shallow aquifers (Tertiary Post-Rupelian, Quaternary 2, and Quaternary 1) is plotted for the High, Medium, and Low scenarios. The values (crosses) are compared to the mass of salt that would be displaced without the injection (circles) due to the base flow of salt towards the rivers. The base flow is almost the same over each layer for each specific scenario. This shows that for each scenario a different quasi-stationary state has evolved. The magnitude of the injection-induced increase in the cumulative salt mass depends on the considered layer. The increase is highest for the flow into the Tertiary Post-Rupelian,

because of the high concentrations that are found on the top of the Rupelian clay barrier (bottom of Tertiary Post-Rupelian), and it decreases strongly across the Quaternary 2 and Quaternary 1. Therefore, the magnitude of the concentration increase after the injection strongly depends on the salt distribution prior to the injection, or in other words: a notable increase in concentration will most likely only occur where elevated salt concentrations already exist prior to the injection.





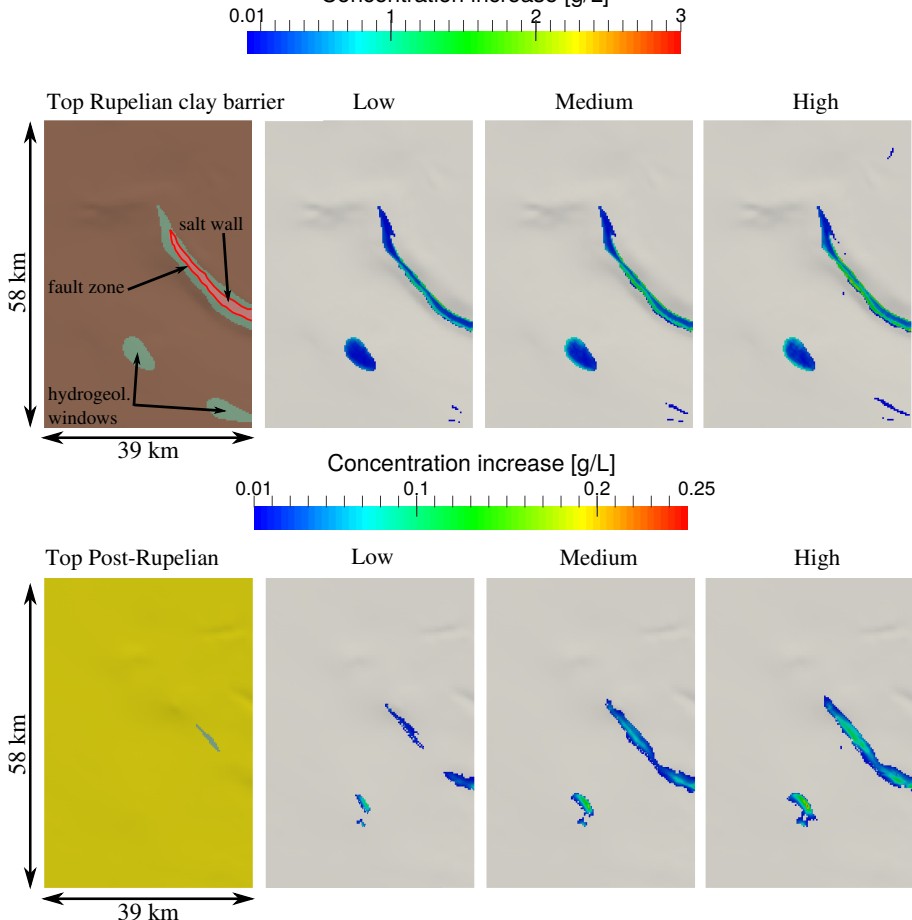

**Figure 9. Top row:** View on top of the Rupelian clay barrier. The top left figure shows the top view on the geology (as shown in Fig. 7 a). The three figures on the right show the salt concentration increase after 50 years of injection, for the three different scenarios Low, Medium and High with increasing initial salt gradients. Concentration increases below 0.01 g/L are not shown. **Bottom row:** View on top of the Post-Rupelian. The bottom left figure shows the top view on the geology of the Post-Rupelian. The three figures on the right show the salt concentration increase after 50 years of injection, again for the same three scenarios. Note the different scales on the legends for the top and bottom row.

## Scenario study 2: Boundary conditions

Next, different types of lateral boundary conditions of the regular domain are compared. Here, the reference scenario employs infinite aquifers as the lateral boundaries. The remaining two scenarios consider a Neumann no-flow and a Dirichlet (hydrostatic) boundary condition. Figure 11 shows the mass flow displaced into the shallow aquifers over the hydrogeological

5   windows (left), and the part which is displaced over the fault zone (right), for the three scenarios. It can be seen that the choice of boundary conditions strongly influences the flow regime in the whole system. For the no-flow scenario, considerably more




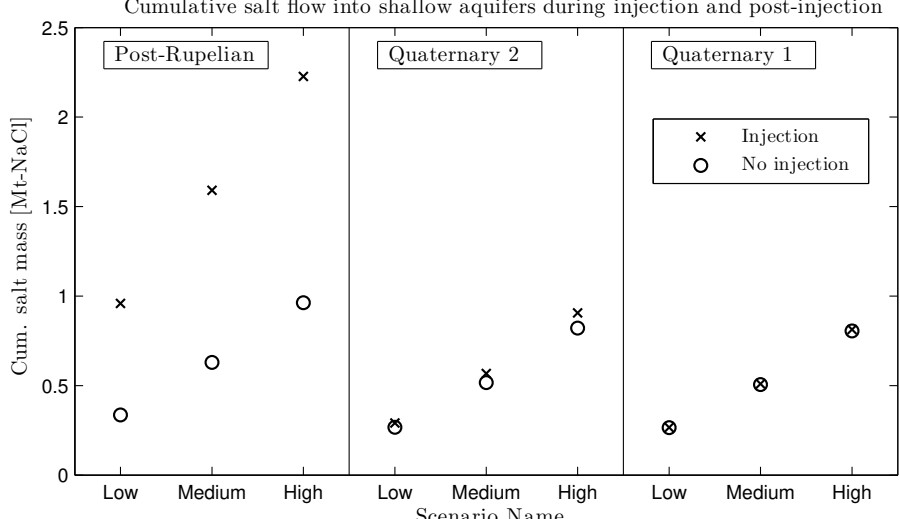

**Figure 10.** The cumulative salt flow for a period of 100 years (injection + post-injection) into the Tertiary Post-Rupelian, the Quaternary 2, and the Quaternary 1 each for the low, medium, and high scenario. The crosses correspond to simulations with injection, whereas the circles correspond to cases without injection, i.e. only the salt base flow.

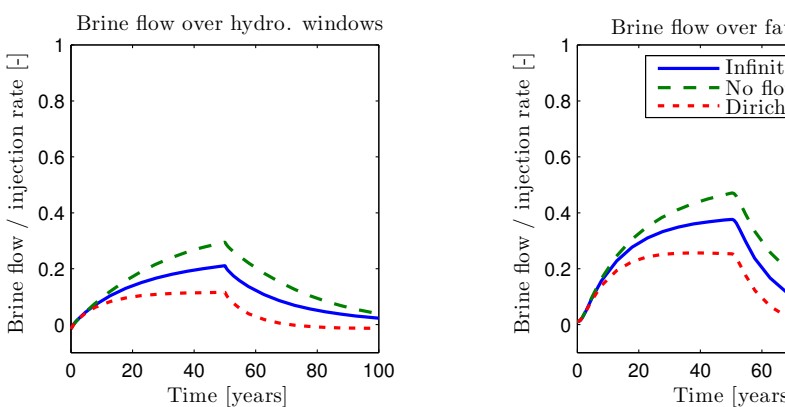

**Figure 11. Left:** brine flow over hydrogeological windows into the shallow aquifers (normalized by the injection rate). **Right:** brine flow over fault zone into the shallow aquifers.

fluid is displaced vertically than for the Dirichlet scenario. The results of the infinite-aquifer scenario fit somewhere in between. This is expected as there is more storage capacity available in the extended aquifers than in the no-flow scenario, and there is a stronger resistance at the lateral boundaries in the infinite aquifer scenario than in the Dirichlet scenario. For the Dirichlet scenario, the leakage rate becomes stationary after 30 years of injection and quickly reduces at the end of the injection. For the infinite-aquifer and the no-flow scenarios, the leakage rate stays elevated even 50 years after the injection has stopped. In





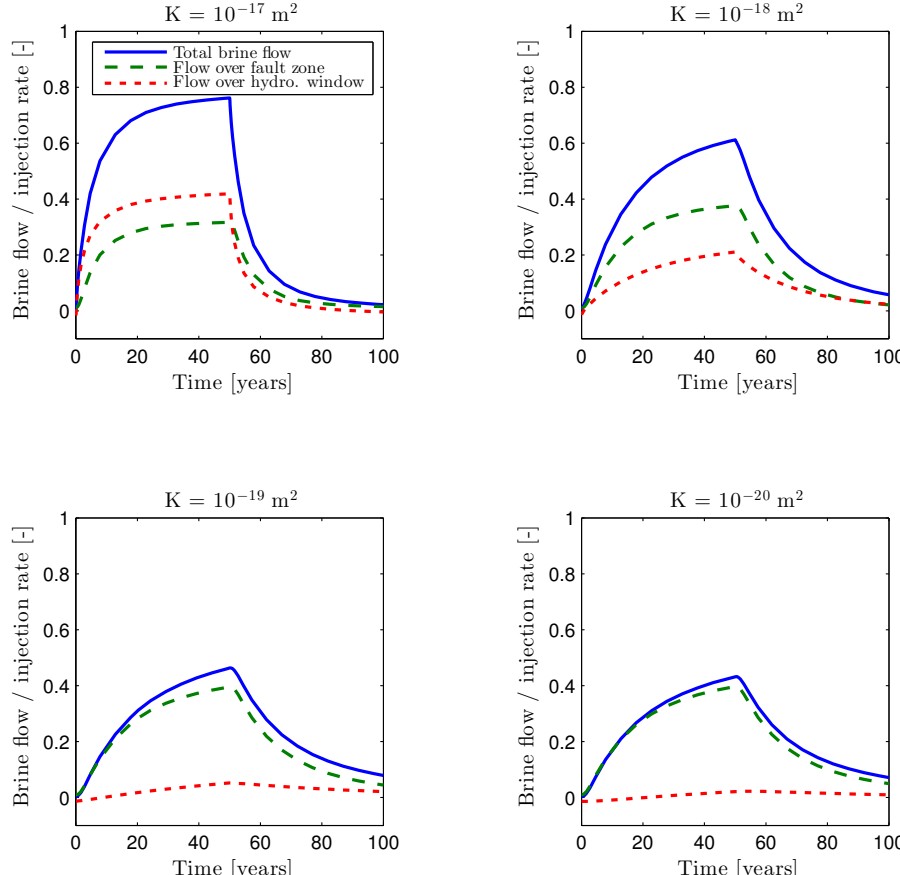

**Figure 12.** Brine flow normalized by the brine injection rate into the shallow aquifers, for different permeabilites of the Upper Buntsandstein barrier. The upper right scenario with a permeability of $1 \cdot 10^{-18}$ m$^2$ corresponds to the reference case.

all three scenarios, the leakage rates into the shallow aquifers reach a significant level compared to the injection rate, which is caused by the Dirichlet boundary conditions prescribed on the top of the geological model at the rivers.

**Scenario study 3: Upper Buntsandstein barrier permeability**

Within this scenario study, the permeability of the layer confining the injection layer, i.e. the Upper Buntsandstein barrier, is varied over several orders of magnitude. The results are presented in Fig. 12. The higher the Upper Buntsandstein barrier permeability, the more diffuse leakage through this barrier will occur. This will also increase the flow through the hydrogeological windows in the Rupelian clay barrier directly above the injection point. The flow field completely changes when the barrier permeability is decreased, and focused leakage through the fault zone becomes the predominant leakage path. The overall amount of displaced fluid into the shallow aquifers decreases with decreasing barrier permeability. Diffuse leakage becomes less important at low barrier-rock permeabilities between $1 \cdot 10^{-19}$ and $1 \cdot 10^{-20}$m$^2$. The simulations show the importance of




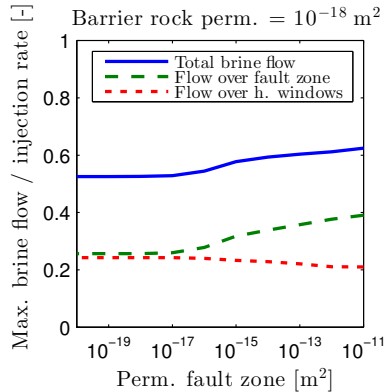 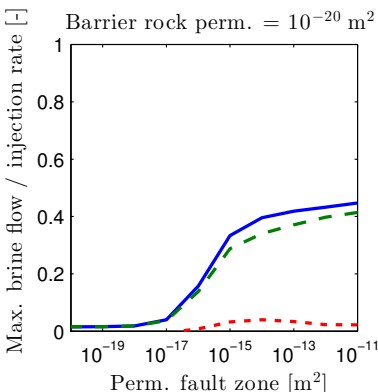

**Figure 13.** Maximum brine flow into the shallow aquifers reached after 50 years of injection (normalized by the injection rate) over the fault zone permeability. **Left:** high Upper Buntsandstein barrier permeability, high diffuse migration through barrier; **right:** low Upper Buntsandstein barrier permeability, high focused migration through fault zone.

the Upper Buntsandstein barrier permeability in controlling diffuse leakage through the barrier and focused leakage through the fault zone. They further show that a high diffuse leakage rate through the Upper Buntsandstein barrier leads to focused leakage in regions where the Rupelian clay barrier is discontinuous, i.e at the hydrogeological windows.

**Scenario study 4: Fault zone transmissivity**

The last scenario study varies the fault-zone transmissivity for two scenarios: (i) A case with high diffuse migration where the permeability of the Upper Buntsandstein barrier rock is similar to the reference case ($1 \cdot 10^{-18}$ m$^2$); (ii) a scenario where the permeability of the barrier rock is low ($1 \cdot 10^{-20}$ m$^2$) and migration mainly occurs through the fault zone. The results are presented in Fig. 13. Although varying the fault zone permeability has a notable effect in locations where diffuse migration is dominant (left figure), the effect is considerably higher for focused migration (right figure), especially for fault-zone per-

meabilities between $1 \cdot 10^{-17}$ and $1 \cdot 10^{-14}$ m$^2$. For higher fault-zone permeabilities, the flow is less sensitive to permeability changes as the resistance of the fault zone becomes small compared to the resistance within the injection layer. The right figure also shows that if neither a diffuse nor a focused vertical pathway up to the shallow aquifers exists, vertical migration does not occur.

**4.3   Model simplification**

The results of the comparison between the four models given in Table 2 are discussed below. The comparison is carried out for two scenarios: A "focused leakage scenario" with a low Upper Buntsandstein barrier permeability ($1 \cdot 10^{-20}$ m$^2$), where the leakage predominantly occurs through the fault zone (similar to the scenario shown in Fig. 12 bottom right) and a "diffuse leakage scenario" with a high Upper Buntsandstein barrier permeability ($1 \cdot 10^{-18}$ m$^2$). In the diffuse leakage scenario, leakage occurs through both the hydogeological windows and the fault zone (similar to the scenario shown in Fig. 12 top right). The



remaining parameters are similar to the reference setup as shown in Table 1 and in Table 3. For the comparison two target variables are chosen: The volumetric flow into the shallow aquifers at different locations (fault zone and hydrogeological windows), and the pressure buildup at the locations M1 and M2, shown in Fig. 7. The injection rate of each of the five models is chosen such that the injected volume is equivalent to the constant injection rate $0.5 \, \mathrm{Mt \, year^{-1}}$ of $CO_2$ under the initial

conditions at the injection point.

**Focused leakage scenario**

The flow over the vertical pathways and the pressure buildup at the measurement points is given in Fig. 14 for the focused leakage scenario. The top left figure shows that almost no flow over the hydrogeological windows occurs for all models, which

is expected as there is hardly any diffuse migration over the Upper Buntsandstein barrier in this scenario. The top right of Fig. 14 shows that the highest flow for the Analytical Model, followed by the 1p1c model, both methods neglecting variable-density salt transport. This is also reflected in the bottom right plot showing the pressure buildup at M2 located in the fault zone (13.5 km from the injection point). Here, the Analytical and the 1p1c model show the lowest pressure buildup. Both observations, the high flow rates over the fault zone combined with the small pressure buildup at M2, can be attributed to a

lower resistance against flow within the fault zone for the 1p1c and the Analytical Model. In the models considering variable-density salt transport, the weight of the brine column in the fault zone increases as brine with a high salt concentration is pushed upward during the injection. This leads to a higher pressure buildup and a lower leakage rate over the fault zone. The bottom left plot in Fig. 14 shows the pressure buildup at M1, located 6.2 km from the injection point. Here, the 2p3c model shows the smallest pressure buildup compared to the other models. There are two main factors contributing to the small

pressure buildup: (i) the injection-induced volume decrease of the $CO_2$ plume related to the increasing density of $CO_2$ and (ii) the upward movement of the $CO_2$ plume along the anticlinal structure away from the fault zone and therefore also the measurement points M1 and M2 (see Fig. 7 b) for orientation). The pressure buildup of the 1p1c and the 1p2c model in M1 are in very good agreement. The good agreement can only be obtained when treating the salt concentration and the temperature as pseudo components, varying linearly in depth and constant in time, in the 1p1c model. This yields brine viscosities similar to

those of the 1p2c model in the injection horizon.

 When estimating leakage rates, it is important to also take into account the layers of the Middle and Lower Buntsandstein which surround the actual injection horizon, the Solling sandstone. Their thickness (combined 480 m), permeability ($1 \cdot 10^{-16} \, \mathrm{m^2}$), and porosity (0.04) are not negligible (see Table 1). In the Analytical Model, we consider these layers by averaging the values of permeability and porosity of the Lower and Middle Buntsandstein together with the injection horizon (Solling sandstone).

We weight the layer-specific values with their respective thicknesses. For more details see Appendix B. Figure 15 compares the results for the Analytical Model when (i) the injection horizon is comprised of only the Solling sandstone and (ii) when averaging over the whole Lower and Middle Buntsandstein layers. The results show a significantly increased leakage rate for Case (i). This can be explained with the highly increased diffusivity resulting from the reduced pore space available when only considering the Solling layer. These results emphasize the importance of modeling the actual injection layer together with the




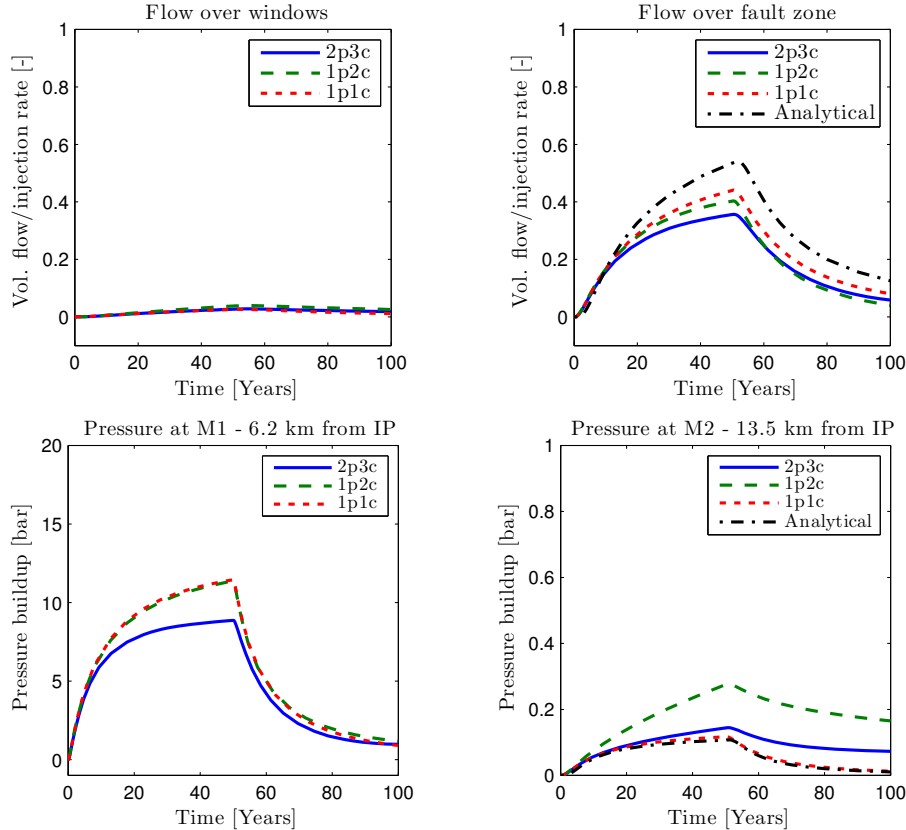

**Figure 14.** Results for the focused leakage scenario (Upper Buntsandstein barrier permeability $1 \cdot 10^{-20}$ m$^2$). **Top row:** Volumetric flow over the hydrogeological windows and the fault zone. **Bottom row:** Pressure buildup at measurement points M1 and M2 in the injection horizon. Note, there are no results for the Analytical Model for the top left plot as the Analytical Model only accounts for flow over the fault zone. Additionally, only the pressure at the fault zone can be determined with the Analytical Model used here.

surrounding overburden and the underlying geological layers for estimating regional-scale brine migration.

**Diffuse leakage scenario**

The detailed results of the diffuse leakage scenario are given in Fig. 16. The Analytical Model is not considered for the diffuse leakage scenario, as it cannot account for diffuse leakage over the barrier. The top left plot of Fig. 16 shows the flow over the hydrogeological windows. The different models show a satisfactory agreement for leakage rates. The highest leakage rates over the fault zone are observed for the 1p1c model and the lowest for the 2p3c model. The pressure buildup in the injection horizon (M1 and M2, bottom row Fig. 16) is lower for the diffuse leakage scenario compared to the focused leakage scenario, as the system's overall resistance to flow is reduced. Again, the 2p3c model shows the lowest pressure buildup at the measurement





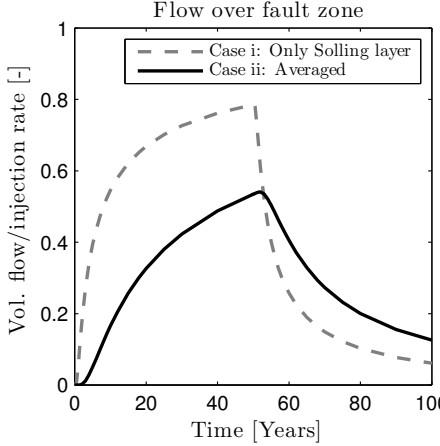

**Figure 15.** Results of the Analytical Model for the two cases: (i) the injection horizon comprises only the Solling sandstone layer and (ii) the injection horizon comprises all layers in the Middle and Lower Buntsandstein with an averaged injection horizon permeability and porosity.

point M1, even declining after 20 years of injection. The explanation is again the injection-induced volume change and the movement of the $CO_2$ plume away from the measurement points, similar to observations for the focused leakage scenario.

Overall, the agreement for the two scenarios (diffuse and focused leakage) between the different numerical models is good. The results of both scenarios show that neglecting variable-density salt transport (1p1c model) does not significantly alter the leakage paths or the pressure buildup in the injection horizon. The large relative differences in the pressure buildup at M2 (at the fault zone) between the different models (ca. factor 2, see Fig. 16, bottom right) need to be viewed in the light of the small absolute values of pressure buildup at M2, which are an order of magnitude smaller than at M1 (for comparison M1: ca. 5-7 bar, M2: ca. 0.1-0.2 bar).

Table 4 compares the results for both diffuse and focused leakage scenarios in terms of the cumulative leakage over 100 years.

## 5 Discussion

All of the presented models can estimate vertical leakage out of an injection horizon over different vertical pathways into shallow freshwater aquifers. The choice of an appropriate model complexity strongly depends on the target variable of interest. For evaluating saltwater intrusion into shallow aquifers it seems reasonable to consider increasing salt concentrations induced by injecting $CO_2$ as a target variable. For this purpose, we use a complex model which accounts for variable-density salt transport and a realistic description of the base flow induced by recharge boundary conditions in the shallow aquifers. We find that two conditions need to be fulfilled in order for notable changes in concentration to occur: (i) the permeability must be high enough such that flow occurs, e.g. where the Rupelian clay barrier is discontinuous, and (ii) initially elevated concentrations need to be present already prior to the injection. The latter implies the need to have good knowledge of the a-priori salt distribution. This is in good qualitative agreement with findings by Tillner et al. (2013), Tillner et al. (2016), Kempka et al. (2013), and Walter et al.





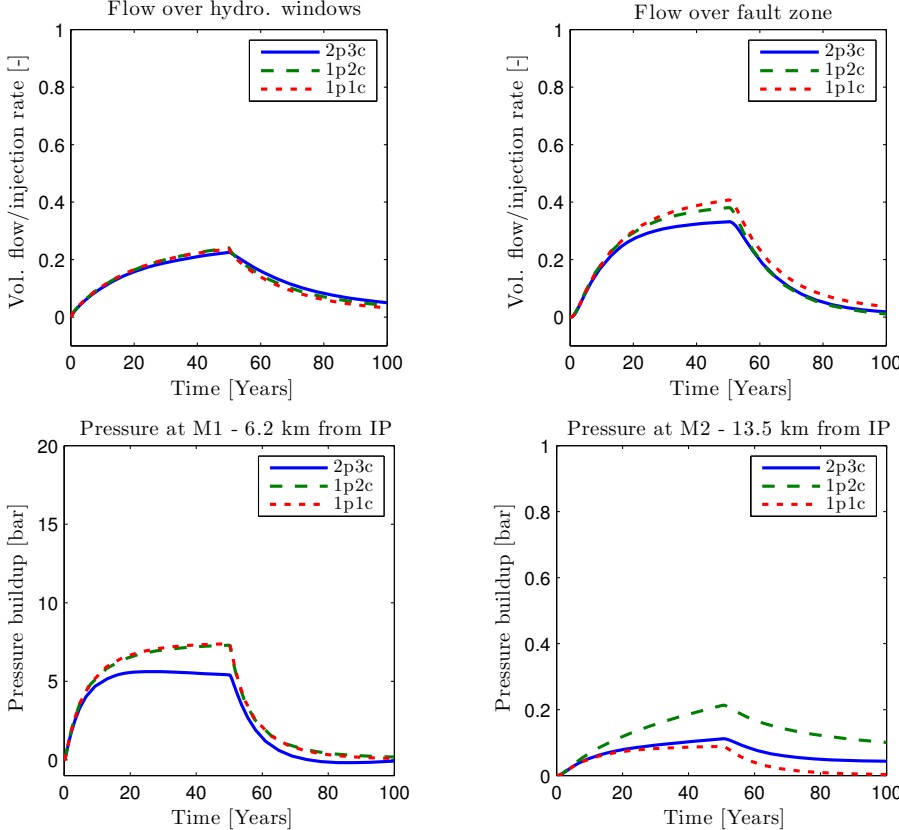

**Figure 16.** High diffuse migration over barrier (Upper Buntsandstein barrier permeability $1 \cdot 10^{-18}$ m$^2$). **Top row:** Volumetric flow over the hydrogeological windows and the fault zone. **Bottom row:** Pressure buildup at measurement points M1 and M2.

(2013). The results also imply that it is unlikely to observe sudden and strong increases in the salt concentration due to $CO_2$ injection at locations where elevated concentrations have not been an issue before. Further, notable changes in concentration occur only locally near pathways where focused leakage occurs.

Determining the initial salt distribution prior to the injection involves large uncertainties in the regional-scale hydrogeological parametrization as well as the establishment of a realistic base flow in the shallow aquifers controlled by recharge boundary conditions. This requires a complex geometry and a computationally demanding model (regional-scale variable-density salt transport). For real sites, this would mean that the model needs to be calibrated against measurements, which are in most cases not readily available.

As mentioned above, notable changes in salt concentration in shallow aquifers require leakage over permeable pathways. Determining leakage rates in terms of mass or volume of brine displaced over vertical pathways can provide valuable information even without actually knowing the salt concentrations. For example, when designing pressure-management strategies for reducing leakage over potential vertical pathways, the leakage volume across a fault zone is a good indicator variable, to optimize




**Table 4.** Fluid volumes displaced into the target aquifers during 100 years of injection over different vertical pathways. The volumes are normalized by the injected volume. Two scenarios are considered: (i) diffuse leakage through the Upper Buntsandstein barrier (UBS) ($k = 10^{-18}$ m$^2$) and (ii) focused leakage (UBS permeability $k = 10^{-20}$ m$^2$). Note that the total volume may be slightly higher than the sum of the volumes displaced over the fault zone and the hydrogeological windows, as a small fraction of the volume is displaced over the intact Rupelian clay barrier.

| Model type | Diffuse leakage scenario | | | Focused leakage scenario | | |
|---|---|---|---|---|---|---|
| | Total volume | Fault zone | Windows | Total volume | Fault zone | Windows |
| 2p3c | 0.67 | 0.39 | 0.24 | 0.48 | 0.44 | 0.01 |
| 1p2c | 0.68 | 0.41 | 0.23 | 0.50 | 0.45 | 0.02 |
| 1p1c | 0.74 | 0.43 | 0.26 | 0.54 | 0.50 | 0.03 |
| Analytical | – | – | – | 0.61 | 0.61 | – |

the placement of wells or brine-withdrawal rates (Birkholzer et al., 2012). Similar reasoning applies to a risk assessment during a site-selection process, where high data uncertainty is addressed with Monte-Carlo methods. If the target variable is expressed in terms of a leakage rate, simplified models are useful, which are quick to set up, have a small data demand, and a reduced computational effort. These models can be considered in addition to or as an alternative to a complex model. An important

question is, how far can the model complexity be reduced before the models become too simple to determine leakage rates. For this reason, we compared the results of different models with varying complexity and discussed the key parameters that control leakage rates into shallow aquifers and pressure buildup in the injection horizon.

A key aspect for the injection of $CO_2$ is the definition of realistic boundary conditions. Dirichlet conditions at the lateral boundaries during the injection lead to underestimation of vertical brine migration. Incorporating no-flow boundaries within

the inner domain, or extending the model to obtain infinite-aquifer boundary conditions allow more vertical brine flow as shown in Fig. 11. If the top boundary above the shallow aquifers were considered to be a no-flow boundary, brine flow into the target aquifers would be significantly smaller, similar to the results found by Walter et al. (2012, 2013), or Cihan et al. (2013), where the displaced brine distributes more into the intermediate aquifers.

The results presented here for diffuse migration across barriers show, that an increased permeability of the Upper Buntsand-

stein barrier leads to more vertical leakage, because the overall vertical resistance decreases. Significant diffuse migration across the barrier changes the flow regime in the intermediate layers (Cretaceous, Pre-Rupelian), resulting in focused migration in locations where the Rupelian clay barrier is discontinuous (hydrogeological windows), even if the Upper Buntsandstein barrier below is intact (see Fig. 12). Diffuse vertical migration is found to be significant for barrier permeabilities higher than $1 \cdot 10^{-19}$ m$^2$, which is in good agreement with findings in Birkholzer et al. (2009). In this work, we simplified main geological

units of the North German Basin, as we assign homogeneous and isotropic permeability values to each layer. However, inter-calated lithological differences within the geological units may reduce the overall vertical permeability. High permeabilities of



the Upper Buntsandstein barrier of $1 \cdot 10^{-18}$ m$^2$ to $1 \cdot 10^{-17}$ m$^2$ are considered to be unlikely.

Varying the fault-zone transmissivity shows that the injection-induced upward flow is most sensitive to this parameter when diffuse migration is small, and the resistance against flow is similar between the injection point and the fault zone as well as over the length of the fault zone. The sensitivity of the leakage rate with respect to the fault-zone transmissivity is small for

fault-zone permeabilities higher than $(1 \cdot 10^{-14}$ m$^2)$, even when increasing the fault-zone permeability over several orders of magnitude. Therefore, for the case of a highly permeable fault zone, a simplified geometrical representation of the fault zone, as used in this work, is considered sufficient.

The layers surrounding the injection horizon in our geological model are the Lower Buntsandstein and the Middle Buntsandstein. They have a combined thickness of 500 m, while the injection horizon itself, i.e. the Solling sandstone, has a thickness

of only 20 m, and a permeability of $1 \cdot 10^{-16}$ m$^2$. Their contribution to the overall storage of displaced brine is significant, causing a strong reduction of vertical flow during the injection period (see Fig. 15). Another important parameter influencing vertical leakage rates, which is not considered here, is the rock compressibility. This was already discussed in detail in Schäfer et al. (2011).

Models using different simplifying assumptions are compared in this work. Injecting a volume-equivalent rate of brine (1p2c

model) instead of CO$_2$ (2p3c model), thereby neglecting the two-phase flow region near the injection well, leads to slightly increased leakage volumes into the shallow freshwater aquifers and to a reduced pressure buildup in the injection horizon. Both effects are related to the injection-induced volume decrease of CO$_2$ and the movement of the CO$_2$ plume along the anticlinal structure away from the fault zone and the pressure measurement locations (M1 and M2). With regard to the distribution of flow among the vertical pathways (hydrogeological windows and fault zone), the 1p2c and 2p3c models are in good agreement.

The injection of brine instead of CO$_2$ can thus be considered a reasonable conservative assumption that simplifies the model considerably. This assumption has also been previously discussed in the literature for example by Cihan et al. (2013).

All models neglecting variable-density salt transport (1p1c and Analytical Model) show an increased leakage rate over the fault zone and a decrease in pressure buildup near the fault zone. This behavior is related to highly saline water being pushed upwards along the fault zone, which increases the weight of the vertical fluid column. However, leakage rates for the 1p1c

model compared to the 1p2c model do not reflect this. A new hydrostatic equilibrium, which means that the injection-induced flow over the fault zone ceases entirely during the injection period due to the increasing gravitational force, such as discussed in Oldenburg and Rinaldi (2010), is not observed in any of the simulations in this work.

The Analytical Model presented by Zeidouni (2012) relies on the assumption of perfectly horizontally stratified layers. Further, it does not account for diffuse leakage through the barrier layers. Thus, it is not suitable for our diffuse leakage scenario and

should only by applied to scenarios with predominant focused leakage. The highest vertical leakage over the fault zone is observed with this model. The volume displaced into the shallow freshwater aquifers after 100 years is 27% higher than for the 2p3c model (see Table 4). However, this overestimation of leakage by the Analytical Model comes at almost negligible computational costs. The Analytical Model is therefore a useful tool to quickly assess the consequences of changing certain parameters within the geological model or to obtain conservative estimates on leakage rates over the fault zone. An alternative





analytical solution is presented in Cihan et al. (2011), which is also capable of handling diffuse migration over barrier layers. Applying this model to the case study presented here, would increase the applicability range of analytical solutions.

## 6 Conclusions

The main findings of this work are summarized below:

– Notable, in the sense of non-negligible, increases in salt concentration in the target aquifers are locally constrained to regions, where initially elevated concentrations are present prior to the injection, and where permeabilities are high enough to support sufficient flow. Hence, the quality of the prediction of concentration changes strongly depends on how well the initial salt distribution is known.

   – An inherent problem to modeling is the assignment of boundary conditions. Lateral and top boundary conditions strongly
determine the amount of displaced brine into the target aquifers. Lateral Dirichlet boundary conditions at insufficient distance from the injection will lead to a strong underestimation of vertical flow. Setting the top boundary condition as *open* - as opposed to a closed boundary at the top - strongly increases the amount of fluid that is displaced into the target aquifers.

   – The permeability of the Upper Buntsandstein barrier plays a crucial role in determining the amount of diffuse leakage.
Diffuse migration through the Upper Buntsandstein barrier can result in focused leakage in locations where the Rupelian clay barrier is discontinuous.

   – For a realistic estimate of vertical leakage, the storage potential for displaced brine of the Lower and Middle Buntsandstein surrounding the injection horizon Solling needs to be considered.

   – During injection (assuming a constant injection rate), the consideration of variable-density salt transport decreases the
overall vertical leakage and increases the pressure at the fault zone due to the additional resistance caused by the increase in the gravitational force. However, the effect of variable-density salt transport on the leakage rates does not seem to be significant.

   – Injecting an equivalent volume of brine, instead of $CO_2$ is a conservative assumption which leads to slightly increased brine flow into the shallow aquifers and a reduced pressure buildup in the injection horizon.

– The Analytical Model considered in this work shows the highest focused leakage rates over the fault zone, as it does not consider variable-density salt transport or diffuse leakage over the barrier layers. Despite of the overestimation of leakage, it is a very useful tool for a quick and conservative assessment of leakage rates.





**Data availability**

In order to obtain the simulation code DuMu$^x$ (Version 2.8.0; Schwenck et al. (2015)) has to be installed along with the DuMu$^x$-Pub module[1] containing the problem setup and grids. For further information on the installation of DuMu$^x$ please visit the dumux homepage[2] and look into the README in the DuMu$^x$-Pub module.

**Appendix A: Numerical model: Balance equations and solution method**

In this work, three model types which differ with respect to the number of components and phases, are compared. The most complex model is a two-phase three-component model (2p3c):

$$\frac{\partial(\phi \sum_\alpha \varrho_\alpha^{mol} x_\alpha^\kappa S_\alpha)}{\partial t} - \sum_\alpha \nabla \cdot \left\{ \varrho_\alpha^{mol} x_\alpha^\kappa \frac{k_{r\alpha}}{\mu_\alpha} \mathbf{K}(\mathbf{grad}\, p_\alpha - \varrho_\alpha \mathbf{g}) + \varrho_\alpha^{mol} \mathbf{D}_{\alpha,\mathbf{pm}}^\kappa \mathbf{grad}\, x_\alpha^\kappa \right\} = q^\kappa, \tag{A1}$$

$$\alpha \in \{w,n\} \text{ and } \kappa \in \{H_2O, CO_2, NaCl\},$$

where the phase index $\alpha$ represents the phases: wetting (w, brine) and non-wetting (n, $CO_2$). The component index $\kappa$ represents the components water ($H_2O$), carbon dioxide ($CO_2$) and salt ($NaCl$). $\phi$ is the effective porosity, $\varrho_\alpha^{mol}$ is the molar and $\varrho_\alpha$ the mass density of phase $\alpha$. $x_\alpha^\kappa$ is the molar fraction of component $\kappa$ in phase $\alpha$, $S_\alpha$ is the saturation, $k_{r\alpha}$ is the relative permeability, $\mu_\alpha$ is the dynamic viscosity, $\mathbf{K}$ is the intrinsic permeability tensor, $p_\alpha$ is the phase pressure, $D_{\alpha,pm}^\kappa$ is the effective diffusion coefficient of the porous medium. The model can account for miscibility of the two phases, however we are not

primarily interested in the fate of the injected $CO_2$, therefore we consider the two phases immiscible. Salt is only present in the brine phase. In conclusion, the wetting phase consists of the components water and salt and the non-wetting phase only of $CO_2$. Neglecting the effects of two phase flow we arrive at a single-phase, two-component model (1p2c):

$$\frac{\partial(\phi \varrho_w^{mol})}{\partial t} - \nabla \cdot \left\{ \varrho_w^{mol} \frac{\mathbf{K}}{\mu_w} (\mathbf{grad}\, p_w - \varrho_w \mathbf{g}) \right\} = q_w, \tag{A2}$$

$$\frac{\partial(\phi \varrho_w^{mol} x_w^{NaCl})}{\partial t} - \nabla \cdot \left\{ \varrho_w^{mol} x_w^{NaCl} \frac{\mathbf{K}}{\mu_w} (\mathbf{grad}\, p_w - \varrho_w \mathbf{g}) + \varrho_w^{mol} \mathbf{D}_{w,pm}^{NaCl} \mathbf{grad}\, x_w^{NaCl} \right\} = q^s, \tag{A3}$$

where Eq. A2 is the total mole balance of brine and Eq. A3 is the transport equation for NaCl. Further neglecting the effects of variable-density flow due to salt transport leads to a single-phase, single-component model (1p1c) where salt is considered as a pseudo component which influences the brine viscosity and the density (similar to a constant geothermal gradient) but the salinity stays constant during the simulation.

In all models the porosity is a function of pressure under the assumption of a constant compressibility:

$$\phi = \phi_{ref} \left( 1 + X + \frac{X^2}{2} \right), \ X = C(p - p_{ref}). \tag{A4}$$

Here $\phi_{ref}$ is the reference porosity, $C$ is the compressibility and $p_{ref}$ is the reference or initial pressure. The equations of state used in the models are given in Table A.1. For spatial discretization, the BOX-Method is used, which is a node-centered finite

---

[1]https://git.iws.uni-stuttgart.de/dumux-pub/Kissinger2016a.git
[2]www.dumux.org





**Table A.1.** List of equations of state used for calculating the fluid properties of $CO_2$ and brine as well as relationships for capillary pressure and relative permeability. The salinity is defined here as the mass fraction of salt.

|  | Symbol | Unit | Function of ... | Reference |
|---|---|---|---|---|
| Density $CO_2$ | $\varrho_n$ | kg m$^{-3}$ | $f(p,T)$ | Span and Wagner (1996) |
| Dynamic viscosity $CO_2$ | $\mu_n$ | Pa s | $f(p,T)$ | Fenghour et al. (1998) |
| Density brine | $\varrho_w$ | kg m$^{-3}$ | $f(p,T,\text{salinity})$ | Batzle and Wang (1992) Adams and Bachu (2002) |
| Dynamic viscosity brine | $\mu_w$ | Pa s | $f(T,\text{salinity})$ | Batzle and Wang (1992) Adams and Bachu (2002) |
| Capillary pressure | $p_c$ | Pa | $f(S_n)$ | neglected |
| Relative permeability | $k_r$ | - | $f(S_n)$ | Brooks and Corey (1964) |

volume method based on a finite element grid, see Helmig (1997) for further reference. For temporal discretization, a fully implicit scheme, using the Newton method to handle the system of non-linear partial differential equations, is applied.

**Fault zone representation using discrete fracture model**

We consider a fault zone with a width of 50 m, while the horizontal discretization length is about 300 m. Thus, representing
the geometry of the fault zone accurately would require grid refinement over large areas, which would drastically increase the computational costs. To avoid refinement the fault zone is modeled with a discrete fracture approach. The fractures are defined on the element faces, which leads to a simplification of the geometry but avoids a severe grid refinement. Nodes connected by a fracture consider both matrix and fracture flow. Fracture flow consists of only advective flow (no diffusive flow) using a two-point flux approximation. Storage in the fracture is considered with an additional storage term for nodes connected to
a fracture. A fracture can be described by three parameters: fracture width, fracture permeability, and fracture porosity. The position of the fault zone is illustrated in the schematic shown in Fig. 5 marked in red.

**Appendix B: Further information on the Analytical Model**

In this study, we apply the analytical solution presented in Zeidouni (2012) for single-phase brine injection in a horizontally stratified system of aquifers which are coupled through a permeable fault zone. The Analytical Model considers the barrier
layers as completely impermeable. We have adapted the geological model presented in Sec. 2 to the Analytical Model (see Fig. B.1). The shallow aquifers (Post-Rupelian, Qaternary 2 and Quaternary 1) act as a Dirichlet boundary condition, which is



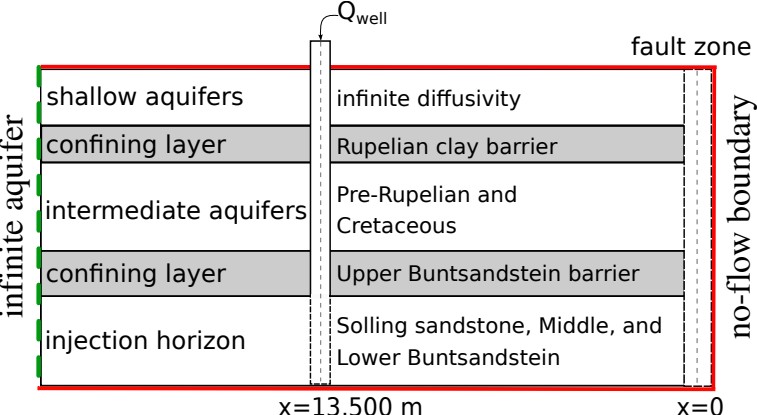

**Figure B.1.** Model setup for the Analytical Model with two permeable layers (injection horizon and intermediate aquifers). and the shallow aquifers with infinite diffusivity.

**Table B.1.** Values for the permeability, porosity, viscosity and diffusivity of the injection horizon, the intermediate aquifers, and the shallow aquifers for the Analytical Model. For the injection horizon, we differentiate between the two cases shown in Fig. 15.

| Parameter | Unit | Case i: Injection horizon (Solling only) | Case ii: Injection horizon (averaged) | Intermediate aquifer (averaged) | Shallow aquifer (infinite diffusivity) |
|---|---|---|---|---|---|
| Thickness | m | 20 | 500 | 1250 | – |
| Permeability | $m^2$ | $1.1 \cdot 10^{-13}$ | $4.5 \cdot 10^{-15}$ | $3.5 \cdot 10^{-14}$ | – |
| Porosity | – | 0.2 | 0.046 | 0.078 | – |
| Viscosity | Pa s | $6.7 \cdot 10^{-4}$ | $6.7 \cdot 10^{-4}$ | $7.5 \cdot 10^{-4}$ | – |
| Total compressibility | $Pa^{-1}$ | $9 \cdot 10^{-10}$ | $9 \cdot 10^{-10}$ | $9 \cdot 10^{-10}$ | – |
| Diffusivity | – | 0.913 | 0.161 | 0.664 | $\infty$ |

achieved by setting the combined diffusivity ($D = \frac{K}{\mu \phi c_t}$, $c_t$ is the total compressibility) of these aquifers to a very high value. The actual injection horizon, Solling sandstone (thickness 20 m), is embedded between the Middle Buntsandstein (thickness 130 m) and the Lower Buntsandstein (350 m) which both have a permeability of $1 \cdot 10^{-16}$ $m^2$ and a porosity of 0.04, see Table 1. These layers contribute to the storage potential and therefore decrease the diffusivity of the injection horizon. Hence,

5 permeability and porosity of the injection horizon are recalculated for the Analytical Model by taking an arithmetic average of the three layers, weighted by their specific layer thicknesses (equivalent to Case ii in Fig. 15). Similarly, the permeability and porosity of the intermediate Cretaceous and Pre-Rupelian layers are averaged to obtain one layer. The resulting parameters are given in Table B.1. The viscosity of the aquifers is estimated from temperature and salinity conditions at the relevant depths.





*Author contributions.*

| Holger Class and Alexander Kissinger | Numerical modeling |
| Stefan Knopf and Vera Noack | Geological expertise and geological model setup |
| Wilfried Konrad and Dirk Scheer | Participatory modeling |

*Acknowledgements.* This study is part of the CO2BRIM research project. A major goal of the project is introducing participatory modeling in a joint engineering and social science approach as a means to involve potential stakeholders of $CO_2$ storage applications into the technical modeling process. The authors gratefully acknowledge the funding for the CO2BRIM project (03G0802A) provided by the German Federal Ministry of Education and Research (BMBF) and the German Research Foundation (DFG) within the geoscientific research and development program Geotechnologien.

Additionally, we would like to thank Christoph Jahnke for fruitful discussions and very helpful advice.





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
