# Peer review of "Regional-scale brine migration along vertical pathways due to $CO_2$ injection – Part 2: a simulated case study in the North German Basin"

_Hydrology and Earth System Sciences, 2016_

## Referee Comment (RC1) · Anonymous Referee #1 · 8 Feb 2017

This manuscript addresses a very relevant topic, which should be of general interest to HESS readers. Novel approaches are presented and substantial and clearly justified conclusions are reached. The methods used are stated completely and clearly (with my more specific remarks to be considered below). I think, some more discussion on individual findings would improve the paper even further. I give some statements concerning this below, which are intended to assist the authors in this point. The abstract and title reflect the type of work and findings, the paper is well written and structured. The figures are required (one exception listed below) and of good quality. The paper contains lots of interesting findings, which allow for a better system understanding of the model considered. Overall, I think this is a good paper very suitable, and with

the modifications indicated below this should be a valuable contribution to the HESS journal.

- Vertical exaggeration too small, I do not see topography

- The main feature is the fault zone and the corresponding parameters. How is this connected to paper 1? Is this the end of the expert discussions? Why such a high value? This is higher than most saline formations...

- Mesh too coarse at fault zones. Any grid resolution study? Have you done a grid effect study to test if you capture the right effects? What did you find?

- Justify the reference model. It is missing thermal effects, which provide additional buoyancy.

- Fig. 8: 5 Isolines instead of 6? Please correct number.

- Describe infinite aquifer BC better. I think it is closed, but at 100 km extra (see also comment further below)

- Page 18, line 10 ? I do not see this in the Figure. Please be clearer here.

- The upper boundary condition drives the brine through the fault and hydrogeological window into the shallow aquifers and leads to high salinization (up to 50% of injected brine) in the reference model that is considered as the most likely geological setup (p 12. line 6f). This high leakage rate actually appears as unlikely and already an effect of the BC. Please comment on this a bit more in you manuscript.

- Give Scenario in Figure captions and state varied variable more clearly, that makes it easier to read.

- Sensitivity for Fig. 15 should be calculated using the 2p3c or 1p2c model, not the simplification, which shown to yield wrong results in Fig. 14. Why the model comparison anyway.

[Figure]

- Better scaling of figures (Fig. 14 top left, Fig. 11, Fig. 13, Fig. 16)

- Fig 5 should be drawn using the model area, to indicate the BC clearly. Where are the BC at the salt wall in the 3D model?

- I can see differences in Pressure and fluid mass in 4.2. Please also show salt mass and concentration changes, because that is what is interesting from the point of view of the near surface aquifers. And here, the simulators should be different.

- Make a better connection to the first part of the paper also for the parameters and processes assumed important. Was the model complexity also discussed? What was the outcome of the "expert" discussions?

- Clarify maximum salinity in Table 3. kg_NaCl / kg_brine or kg_NaCl / l_brine ?

- Also concerning the initial salt gradient: I am wondering how the salt gradient at the beginning of the spin-ups can influence the initial salt distribution in the scenario study 1, which is a balance between the impacts of diffusion and gravity? Is there an upper boundary conditions for salt in the spin-ups?

- Scenario study one: Better state Low/medium/High salinity gradient instead of just low/medium/high.

- Show Fig 10 as table. Instead, a plot of the isolines for the different cases like in Fig. 8 would be very interesting, to see the changes. Maybe for one or two cases this should be possible.

- Discussion section: I do not agree that leakage rate is a good indicator. For modellers this is true, but how do you measure and how do you detect? Additionally, as you state yourself, the salt concentrations may be very different, so NaCl-mass flow rates would also be of interest. The same volume of leaked water has a very different impact with high concentration brine as compared to low concentration brine.

- I am unhappy about the term "infinite aquifer" boundary, because obviously that is not

the case. The Dirichlet case is that one, the "infinite aquifer" case has just a higher volume and thus dampens the pressure response by a larger overall compressibility. The assessment of the BC is important, I agree, but just assuming that 100 km are "infinite" is wrong, as Figure 11 shows. Please use different terms, and better justify the assumption on your BCs. Check e.g. Benisch and Bauer (2013) for an estimation of the connected boundary volume, which can be appended to the model as bulk storage, to represent this case better. These authors actually also found that the low permeability sections contribute just as much to compressibility as the aquifers, so storage is omitted from the model by extending just the aquifers.

- Discussion: Please do not just cite the Schäfer paper, but also state their findings. This is interesting here.

- Your paragraph on the model comparison and leakage rates (Page 24, line 22 ) is confusing. Please state more clearly, that the simple model do not capture the leakage rates correctly, as they do not transport salt. The hydrostatic equilibrium is a function of the BC and of the injection rate in your case. Compare Delfs et al (2016) for this effect – you probably would need to include temperature for this in the simulations and look at the Dirichlet case.

- "The Analytical Model is therefore a useful tool to quickly assess the consequences of changing certain parameters within the geological model or to obtain conservative estimates on leakage rates over the fault zone" Have you done more than the two model comparison runs to justify this statement? For one model run, the analytical model does not work, for the other you get a 30% overestimation. A comparison for a range of parameters as in the parameter study would be helpful, to assess the capabilities of the analytical model fully. However, I do agree that these models are very useful. However, because they have many limitations, a clear understanding of these limitations is helpful (but maybe also beyond scope here).

- I agree with the conclusions, but they are highly repetitive with the discussion section.

How about merging these sections?

---

## Referee Comment (RC2) · Anonymous Referee #2 · 22 Feb 2017

Regional-scale brine migration along vertical pathways due to CO2 injection – Part 2: a simulated case study in the North German Basin

By Alexander Kissinger et al.

I was involved already as a reviewer in the first round of reviewing. I find the applied changes of the manuscript in accordance to both reviewer comments suitable.

Below are some comments about the revised manuscript:

First of all, the text is still for my impression far too long and the authors should try to shorten it where possible.

A linkage with the participatory modeling approach (part 1) is missing at all. This should be changed.

The motivation to identify a setup with a minimum of computational cost while delivering a maximum of valuable results is understandable. However, it should be emphasized that every simplification leads to a loss of accuracy. If this is kept in mind it is absolutely okay to use these somewhat simplified models for understanding the occurring processes and for obtaining estimates.

I do understand that the study shows that it seems to be okay to neglect buoyancy effects for this model. However, for this finding maybe more simple box model studies (with comparisons to experiments) would be a more straight forward approach. Also, the numer of simplifications is quite accumulated, keeping in mind that the effect of $CO_2$ (two phase flow) is neglected in the first.

The conclusions are mainly banal.

Buoyancy effects due to temperatrue are not discussed. Peclet numbers (temperature and concentration) and Rayleigh numbers (occurrence of free convection) are not discussed.

The study is about injecting $CO_2$ into the middle Buntsandstein in the North German Basin. However, it is well known that the Buntsandstein is under very high formation pressure (more than 600 bar). Have the authors considered the extreme effort of pumping $CO_2$ in to such an overpressurized formation, for instance by considering the energy consumption?

Page 25, Line 7 – 8, this sentence seems to be wrong

—————————

---

## Author Response (AR1)

We thank both reviewers and the editor for their valuable comments. We reply to these comments below:

**Answers to RC1 comments:**

**RC1:** Vertical exaggeration too small, I do not see topography

**Answer:** It is true the topography cannot be recognized in Fig. 2, since there is only a difference of 16 m. Therefore, we show it in more detail in Fig. 6. We make reference to Fig. 6 on p.7 line 19 of the revised version.

**RC1:** The main feature is the fault zone and the corresponding parameters. How is this connected to paper 1? Is this the end of the expert discussions? Why such a high value? This is higher than most saline formations. . .

**Answer:** It is true that a connection to Part 1 is missing here; it was added on p.8 lines 7 and 12. A result of the PM process was to use different transmissivities for the fault zone, which we compared in Fig. 13.

**RC1:** Justify the reference model. It is missing thermal effects, which provide additional buoyancy.

**Answer:** We considered the temperature through a constant geothermal gradient which influences the fluid properties like viscosity and density. We agree that thermal effects resulting from heat transport could play an important role in determining the salinity distribution prior to the injection, especially in regions near the Rupelian Clay barrier where the salinity is still relatively low and temperature effects may dominate (e.g. cold water moving below the Rupelian clay barrier see for example Noack et al. 2013, Kaiser et al. 2013). We discuss this issue in the discussion part of the paper, p.23 lines 10f.

**RC1:** Fig. 8: 5 Isolines instead of 6? Please correct number.

**Answer:** Corrected in revised version.

**RC1:** Describe infinite aquifer BC better. I think it is closed, but at 100 km extra (see also comment further below)

**Answer:** Described in more detail in revised version, also see comment further below.

**RC1:** Page 18, line 10 ? I do not see this in the Figure. Please be clearer here.

**Answer:** We rephrased these sentences in the revised version.

**RC1:** The upper boundary condition drives the brine through the fault and hydrogeological window into the shallow aquifers and leads to high salinization (up to 50% of injected brine) in the reference model that is considered as the most likely geological setup (p 12. line 6f). This high leakage rate actually appears as unlikely and already an effect of the BC. Please comment on this a bit more in you manuscript.

**Answer:** Actually the text on page 13 line 4f states:

*„All scenarios are evaluated against a reference model. The reference model is **not understood** as the most likely geological setup, but simply shows all processes under investigation on a recognizable scale."*

We do not consider this setup to be the most likely, on contrary on page 7 line 23f we state on this matter:

*„Making a conservative assumption, we assume a permeable vertical pathway along the whole flank of the salt wall and refer to it as a fault zone. This fault zone is a permeable connection between the injection horizon and the shallow aquifers above the Rupelian clay barrier. The assumption of fluid migration via vertical pathways in sediments flanking salt structures is a matter of debate. After LBEG (2012) "the contact zone between salt domes and the $CO_2$-sequestration horizon is assumed to be a zone of weakness, similar to geological faults". Such zones of weakness may provide effective vertical migration pathways.*
*To our understanding, the assumption of a permeable fault zone along the whole flank of a diapir is an exaggeration of real geological conditions. In contrast, faults of smaller range at shallower depths in the sediments on top of the hanging wall of diapirs may provide pathways for fluids."*

**RC1:** Give Scenario in Figure captions and state varied variable more clearly, that makes it easier to read.

**Answer:** Done in a revised version.

**RC1:** Sensitivity for Fig. 15 should be calculated using the 2p3c or 1p2c model, not the simplification, which shown to yield wrong results in Fig. 14. Why the model comparison anyway.

**Answer:** With this figure we wanted to show the importance of including the overburden for estimating far-field pressure propagation. The overburden is not only of importance with regard to diffuse leakage that may occur, but also in the focused leakage scenario the overburden plays a crucial role due to its effect on the injection horizon diffusivity. This effect can be shown most conveniently with the analytical method. The diffusivity decreases by a factor of almost 6 if the overburden is considered as well. We have calculated the diffusivities in Appendix B Table B.1 with the formula given inline on page 28 line 1. We have rephrased parts of this paragraph in the revised version.

**RC1:** Better scaling of figures (Fig. 14 top left, Fig. 11, Fig. 13, Fig. 16)

**Answer:** We deliberately always chose the same scale (normalized between 0 and 1). We believe that this makes the figures more comparable among each other. Otherwise differences which are not very large in absolute terms may be overestimated.

**RC1:** Fig 5 should be drawn using the model area, to indicate the BC clearly. Where are the BC at the salt wall in the 3D model?

**Answer:** We redesigned the figure to match it better with the actual geological model. However, it is still a simplified sketch as not all features and boundary conditions (for example rivers) can be shown within one figure, due to the different scales. We make it clear in the figure that the salt wall is a low permeable feature with a constant high salinity (maximum salinity) assigned to it.

**RC1:** I can see differences in Pressure and fluid mass in 4.2. Please also show salt mass and concentration changes, because that is what is interesting from the point of view

of the near surface aquifers. And here, the simulators should be different.

**Answer: Maybe there is a misunderstanding.** Salt concentration changes and cumulative salt mass are shown in Figures 9 and 10. For the model comparison section we did not show these, as the 1p1c and the Analytical solution do not consider transport as such (except as flow times a constant value). If we would assume a constant concentration at the freshwater-saltwater interface, it would scale with volume flow rates shown in the model comparison section.

**RC1:** Make a better connection to the first part of the paper also for the parameters and processes assumed important. Was the model complexity also discussed? What was the outcome of the "expert" discussions?

**Answer:** Indeed, we did not make many connections to Part 1, as in Part 2 we set the focus on the technical findings and did not want to overload the manuscript with the PM methodology, as in the first draft previously submitted to HESS.
However, we generally agree that there should be a better connection. In the revised version, we made more connections to Part 1 in Sections 2 and 3 of Part 2, where the geological and numerical model is presented to make it clear that these were influenced by Part 1 (see  p.8 lines 7 and 13, p.9 lines 9f, p.9 lines 29f) .

**RC1:** Clarify maximum salinity in Table 3. kg_NaCl / kg_brine or kg_NaCl / l_brine ?

**Answer:** It is  kg_NaCl / kg_brine as given in Table 3.

**RC1:** Also concerning the initial salt gradient: I am wondering how the salt gradient at the beginning of the spin-ups can influence the initial salt distribution in the scenario study 1, which is a balance between the impacts of diffusion and gravity? Is there an upper boundary conditions for salt in the spin-ups?

**Answer:** The initial salt gradient (linear increase of salt with depth) is the initial condition for the initialization run. In the initialization run, a base flow is established in the shallow aquifers by assigning a constant groundwater recharge rate of 100 mm/year. This flow is directed towards the rivers shown in Fig. 6 where a fixed atmospheric pressure is assigned.  The initial linear salt distribution changes due to this base flow and upconing of salt occurs near the rivers, see Fig. 8. A more thorough discussion on parameters and processes influencing the salt distribution after the injection run is given in Kissinger (2016).

**RC1:** Scenario study one: Better state Low/medium/High salinity gradient instead of just low/medium/high.

**Answer:** Done in revised version.

**RC1:** Show Fig 10 as table. Instead, a plot of the isolines for the different cases like in Fig. 8 would be very interesting, to see the changes. Maybe for one or two cases this should be possible.

**Answer:** We think that Fig. 10 summarizes an important conclusion of this work, which is that the amount of salt that is being transported into the shallow aquifers due to the injection is strongly

influenced by the initial salt distribution prior to the injection. Therefore, we would like to keep this plot, as in a table, we think this relation will be less obvious.

**RC1:** Discussion section: I do not agree that leakage rate is a good indicator. For modellers this is true, but how do you measure and how do you detect? Additionally, as you state yourself, the salt concentrations may be very different, so NaCl-mass flow rates would also be of interest. The same volume of leaked water has a very different impact with high concentration brine as compared to low concentration brine.

**Answer:** We agree that different NaCl-mass flow rates will have a very different impact. But a notable leakage rate is a necessary requirement for NaCl mass flow rate to occur. The leakage rate is therefore an important information. The uncertainty in determining a realistic salt distribution prior to the injection is very high as many assumptions have to be made and many data points would be necessary for calibrating such a model in order to obtain reliable results. Another possible way of dealing with such uncertainties could be a Monte-Carlo-like approach with simplified methods and assuming a variety of different concentrations.

**RC1:** I am unhappy about the term "infinite aquifer" boundary, because obviously that is not the case. The Dirichlet case is that one, the "infinite aquifer" case has just a higher volume and thus dampens the pressure response by a larger overall compressibility. The assessment of the BC is important, I agree, but just assuming that 100 km are "infinite" is wrong, as Figure 11 shows. Please use different terms, and better justify the assumption on your BCs. Check e.g. Benisch and Bauer (2013) for an estimation of the connected boundary volume, which can be appended to the model as bulk storage, to represent this case better. These authors actually also found that the low permeability sections contribute just as much to compressibility as the aquifers, so storage is omitted from the model by extending just the aquifers.

**Answer:**
We have rephrased the part describing the boundary conditions p.10 lines 22f.
1. We compared different lengths of the applied domain extension (50, 100 and 150 km) and compared those to the analytical solution by Zeidouni (2012), who assumes a constant pressure at infinite distance from the injection. We have attached a section from the dissertation of Kissinger (2016) describing this comparison at the end of this document. We found that 100 km was a sufficient distance to adequately match the results of the analytical solution where pressure changes are zero at infinite distance from the injection. In contrast to e.g. Benisch and Bauer (2013) we did not increase the volume of the outer cells based on an estimation of the storage volume of the relevant layers in the North German Basin. We simply attached additional cells, which are fully connected with each other, whose volume increases towards the boundary of the extended domain. Thereby, we are able to capture the effect of an increasing lateral area due to an increasing radius in the extended domain.
2. We agree that the low permeability sections have a significant contribution to pressure evolution. Since the regular (unextended) model domain already has a large lateral extent (58x39 km), the influence of the low permeable layers is included for a large distance. Simulation test runs showed that a further increase of the low permeable layers did not significantly change the results.

**RC1:** Discussion: Please do not just cite the Schäfer paper, but also state their findings. This is interesting here.

**Answer:** Done in revised version.

**RC1:** Your paragraph on the model comparison and leakage rates (Page 24, line 22 ) is confusing. Please state more clearly, that the simple model do not capture the leakage rates correctly, as they do not transport salt. The hydrostatic equilibrium is a function of the BC and of the injection rate in your case. Compare Delfs et al (2016) for this effect – you probably would need to include temperature for this in the simulations and look at the Dirichlet case.

**Answer:** We rephrased this paragraph in a revised version to make it more clear.

**RC1:** "The Analytical Model is therefore a useful tool to quickly assess the consequences of changing certain parameters within the geological model or to obtain conservative estimates on leakage rates over the fault zone" Have you done more than the two model comparison runs to justify this statement? For one model run, the analytical model does not work, for the other you get a 30% overestimation. A comparison for a range of parameters as in the parameter study would be helpful, to assess the capabilities of the analytical model fully. However, I do agree that these models are very useful. However, because they have many limitations, a clear understanding of these limitations is helpful (but maybe also beyond scope here).

**Answer:** The analytical solution for Case i), shown in Fig. 15 was only done to demonstrate the importance of the low permeable overburden on the simulation results. We consider the analytical solution for Case ii) to be at least comparable to the numerical simulation results for the „focused leakage" scenario.  There are also other analytical solutions which are capable of describing „diffuse leakage" which may be more appropriate for a larger variety of scenarios. We believe that for these kinds of problems, utilizing a combination of analytical and numerical models can be a good idea. Analytical models could be used for quick estimates or Monte-Carlo runs whereas more complex numerical models could be used for verifying the analytical models.

**RC1:** I agree with the conclusions, but they are highly repetitive with the discussion section. How about merging these sections?

**Answer:** We removed the conclusions and merged it as suggested.

**RC1:** Mesh too coarse at fault zones. Any grid resolution study? Have you done a grid effect study to test if you capture the right effects? What did you find?

**Answer:**
We have used a discrete fracture model for describing the fault zone, therefore we are able to assign a fracture width, permeability and porosity. We have conducted a grid convergence study for discretization sizes and fault zone parameters similar to the problem presented in this paper with the analytical solution by Zeidouni (2012) and found that 300x300 m was sufficient for determining volumetric leakage rates (see attached excerpt of Kissinger 2016, DISS_Kissinger_Discretization.pdf Sec. 5.4.1). This grid convergence study was done without considering variable-density flow effects.

Our horizontal grid resolution  is constant at 300x300 m, while our vertical resolution is smaller and varies between 10 and 160 m depending on the layer. The fault zone has a width of 50 m. This resolution is of course a  compromise between accuracy and computational feasibility. Although we did not perform a grid convergence study for salt transport, we have looked at the effect of salt transport with different fault zone volumes and found it to be not very large during the injection when advective forces dominate, at least when compared to other uncertainties within the

hydrogeological system (permeabilities, porosities, fault zone implementation at a salt wall flank in general).

The effects of salt transport: The implementation of the discrete fracture model is such that per default a part of the degree of freedom's (DOF) volume is from the fracture and part from the matrix surrounding the fracture. We have looked at the effect when we neglect the matrix part of the volume at the DOF. This results in a reduced volume at the DOF and therefore a decreased DOF porosity (only fracture volume is active) and an increased transport velocity of the salt water being pushed upwards by the injection. This volume reduction can be as high as a factor of 6 (depending also on fracture and matrix porosities). We have attached a figure describing this comparison from the dissertation Kissinger (2016), see below. Please note that in the figure the „reference" case is not the reference case in this manuscript but the Wm=1 case corresponds to the reference case in this manuscript. Here, we provide a short summary of the results for more details please consider Kissinger (2016). What we see is that, as expected, the salt transport into the shallow aquifers increases for the reduced porosity case (only fault zone porosity) compared to the default case (matrix and fracture porosity considered at the DOF). The overall volumetric leakage is also reduced, which is a result of the increased weight of the fluid column inside the fault zone, eventually leading to an increased resistance against upward flow. A similar behavior of increasing resistance and lower total mass-flow rates due to brine being pushed upward in an improperly sealed abandoned well is reported by Birkholzer (2011). However, the increase in salt transport and the decrease in volumetric leakage we observe for our scenario are rather small.

[Figure]

**Figure 5.20.: Left:** Volumetric flow normalized by injection rate into target aquifers through the fault zone. **Right:** Salt flow into target aquifers through fault zone.

**Answers to RC2 comments:**

**RC2:** First of all, the text is still for my impression far too long and the authors should try to

shorten it where possible.

**Answer:** This will be difficult to achieve, as from the comments of RC1 we got the impression that a revised version would need to comment on further issues in the discussion. However, we removed/merged the conclusion in favor of the discussion section.

**RC2:** A linkage with the participatory modeling approach (part 1) is missing at all. This should be changed.

**Answer:** Indeed, we did not make many connections to Part 1, as in Part 2 we set the focus on the technical findings and did not want to overload the manuscript with the PM methodology, as in the first draft previously submitted to HESS.
However, we generally agree that there should be a better connection. In the revised version, we made more connections to Part 1 in Sections 2 and 3 of Part 2, where the geological and numerical model is presented to make it clear that these were influenced by Part 1 (see  p.8 lines 7 and 13, p.9 lines 9f, p.9 lines 29f) .

**RC2:** The motivation to identify a setup with a minimum of computational cost while delivering a maximum of valuable results is understandable. However, it should be emphasized that every simplification leads to a loss of accuracy. If this is kept in mind it is absolutely okay to use these somewhat simplified models for understanding the occurring processes and for obtaining estimates.

**Answer:** We discuss the limitations of the simplification in the discussion of the revised version.

**RC2:** I do understand that the study shows that it seems to be okay to neglect buoyancy effects for this model. However, for this finding maybe more simple box model studies (with comparisons to experiments) would be a more straight forward approach. Also, the number of simplifications is quite accumulated, keeping in mind that the effect of CO2 (two phase flow) is neglected in the first.

**Answer:** The point we try to make here is:  if we are interested in a realistic distribution of salt for estimating concentration changes due to CO2 injection, it is very important to know about the baseline salt distribution prior to the injection of CO2. Having a model capable of describing the natural conditions prior to the injection is very much dependent on variable-density transport of salt and possibly also heat. BUT: During the injection phase, our results suggest that the effects of variable-density flow are less relevant as advective forces due to the injection dominate.

**RC2:** The conclusions are mainly banal.

**Answer:** We removed the conclusions.

**RC2:** Buoyancy effects due to temperature are not discussed. Peclet numbers (temperature and concentration) and Rayleigh numbers (occurrence of free convection) are not discussed.

**Answer:** We understand that the effects of heat transport can be an important issue for establishing a natural state of the system prior to the injection or even during the injection phase, although presumably to a lesser extent as advective forces will dominate here. We added a part in the discussion section where the effects of neglecting heat transport are discussed, see p. 23 lines 9f.

**RC2:** The study is about injecting CO2 into the middle Buntsandstein in the North German Basin. However, it is well known that the Buntsandstein is under very high formation pressure (more than 600 bar). Have the authors considered the extreme effort of pumping CO2 in to such an overpressurized formation, for instance by considering the energy consumption?

**Answer:** The statement is about the fact, that initial reservoir pressures in Middle Buntsandstein gas fields in NW Germany are relatively high (over-pressured regime). Based on this fact, the reviewer presumes very high absolute formation pressures of over 600 bar in general for Middle Buntsandstein reservoir rock units in the North German Basin. That is in general not the case, as formation pressures are, of course, depth dependent. Known pressure gradients for Middle Buntsandstein gas fields are about 1.3 – 1.5 bar/10m. Hence, in moderate depths (injection point in our model is ca. 1600 m, for example) no extreme reservoir pressures are expected. It would be part of a site-screening and selection process to choose a suitable site with feasible reservoir conditions. Furthermore, the majority of the Middle Buntsandstein gas fields in NW Germany are located in the Lower Saxony Basin (LSB), which is a part of the North German Basin. The characteristics of those gas fields reflect the geological conditions found in the LSB. This sub-basin represents an inversion play with complex reservoir structures formed during Upper Cretaceous inversion phase. The conditions in the LSB, like e.g. relatively high reservoir pressures in the Middle Buntsandstein, should not be extrapolated to the entire North German Basin in a simple manner.

**RC2:** Page 25, Line 7 – 8, this sentence seems to be wrong

**Answer:** The conclusion, where the referred line was found, has been removed in the revised version.

**Literature:**

Birkholzer, J. T., Nicot, J. P., Oldenburg, C. M., Zhou, Q., Kraemer, S., & Bandilla, K. (2011). Brine flow up a well caused by pressure perturbation from geologic carbon sequestration: Static and dynamic evaluations. *International Journal of Greenhouse Gas Control*, *5*(4), 850-861. DOI: 10.1016/j.ijggc.2011.01.003

Kaiser, B. O., Cacace, M., and Scheck-Wenderoth, M. Quaternary channels within the Northeast German Basin and their relevance on double diffusive convective transport processes: Constraints from 3-D thermohaline numerical simulations. Geochemistry, Geophysics, Geosystems, 14(8):3156–3175, 2013.

Noack, V., Scheck-Wenderoth, M., Cacace, M., and Schneider, M. Influence of fluid flow on the regional thermal field: results from 3D numerical modelling for the area of Brandenburg (North German Basin). Environmental Earth Sciences, 70(8):3523–3544, 2013.

Kissinger, A.: Basin-scale site screening and investigation of possible impacts of CO2 storage on subsurface hydrosystems, 2016. doi: http://dx.doi.org/10.18419/opus-8998.
URL http://elib.uni-stuttgart.de/handle/11682/9015

M. Zeidouni. Analytical model of leakage through fault to overlying formations. Water Resources Research, 48(12):n/a–n/a, dec 2012. ISSN 00431397. doi: 10.1029/2012WR012582. URL http://doi.wiley.com/10.1029/2012WR012582.

[Figure]

**Figure 5.7.: a)** Cross-section of the domain showing the lateral and top boundary conditions as well as the injection well and the fault zone (modified after **?**).
**b)** Top view on the upper aquifer showing the inner domain ($30\,\text{km} \times 25\,\text{km}$) in blue and the domain extension with an outer radius of $100\,\text{km}$ in red. Due to the symmetry of the line connecting the injection and the fault zone only half of the domain is simulated.

**5.4.1. Large-Scale Pressure Propagation During Injection**

The driving force for brine migration during the injection period is the displacement of brine caused by the injected fluid. To adequately capture the far-field pressure buildup and brine leakage through the fault zone, the Zeidouni-Method is used. With this method, a reference solution is obtained for a simple test case comprising two aquifers separated by a completely impermeable layer, and connected by a vertically permeable fault zone. The model setup is illustrated in Fig. 5.7 and the relevant parameters are given in Table 5.2.

For this setting two scenarios are evaluated. The first scenario considers a closed top boundary in the upper aquifer (Neumann scenario), and the second scenario considers the upper aquifer to act as a Dirichlet boundary (Dirichlet scenario). This is achieved by increasing the permeability of the upper aquifer by several orders of magnitude, thereby increasing its diffusivity $D_u$ ($D_u = \frac{k_u}{\mu_u \phi_u C_t}$). Here $C_t$ is the sum of the compressibility of the porous medium and the water compressibility. The numerical model is verified against the analytical solution for different radii of the domain extension, as well as different horizontal discretization lengths. Hexahedral elements are used with a constant vertical discretization length of $50\,\text{m}$. The simulations are named according to their horizontal discretization length and the radius of the outer domain, for example: D300-R100 translates into a horizontal discretization length of $300\,\text{m} \times 300\,\text{m}$ and an outer radius of $100\,\text{km}$.

The results for the comparison of different domain lengths are shown in Fig. 5.8. Here, the leakage rates over the fault zone are plotted over time. The leakage rate is normalized by the

**Table 5.2.:** Input parameters for the analytical and the numerical model.
\* denotes parameters only relevant for the numerical simulation.

| Parameter | Unit | Value |
|---|---|---|
| Injection rate | $\mathrm{kg\,s^{-1}}$ | 10.87 |
| Injection period | year | 50 |
| Injection well fault distance | m | 5000 |
| Water viscosity | Pa s | $1 \times 10^{-3}$ |
| Water compressibility | $\mathrm{Pa^{-1}}$ | $4.5 \times 10^{-10}$ |
| Aquifer permeabilities | $\mathrm{m^2}$ | $1 \times 10^{-13}$ |
| Aquifer porosities | - | 0.2 |
| Porous medium compressibility | $\mathrm{Pa^{-1}}$ | $4.5 \times 10^{-10}$ |
| Aquifer thicknesses | m | 50 |
| Barrier permeability* | $\mathrm{m^2}$ | $1 \times 10^{-25}$ |
| Barrier porosity* | - | 0.001 |
| Barrier thickness | m | 50 |
| Fault permeability | $\mathrm{m^2}$ | $1 \times 10^{-12}$ |
| Fault porosity* | - | 0.01 |
| Fault thickness | m | 50 |

injection rate. The results show that for both the Neumann and Dirichlet scenario, a good

[Figure]

**Figure 5.8.:** Comparison of domain radii for 50, 100 and 150 km. **Left:** Neumann case with the upper aquifer closed on top. **Right:** Dirichlet case where the top aquifer has a constant pressure (very large diffusivity).

agreement with the analytical solution is reached for all radii. The cases for an outer radius

of 100 and 150 km are not distinguishable. The solution for the 50 km case shows a slightly smaller leakage rate. Thus, for the simulations presented below, where the complex geological model is used, an outer radius of 100 km is deemed to be sufficient.

The comparison of the different horizontal discretization lengths, for both the Neumann and the Dirichlet scenario are shown in Fig. 5.9. The results show that all three horizontal

[Figure]

**Figure 5.9.:** Comparison of different horizontal discretization lengths for 150, 300 and 450 m. **Left:** Neumann case with the upper aquifer closed on top. **Right:** Dirichlet case where the top aquifer has a constant pressure (very large diffusivity).

discretization lengths sufficiently approximate the analytical solution. While the solution curves of the 300 m and 450 m discretizations are not distinguishable from each other, the solution for the 150 m case follows the analytical solution more closely. This is likely due to the strongly decreased horizontal discretization length, leading to smaller time steps, whose size is constrained by the convergence of the linear solver. A reason for the numerical simulations not exactly fitting the analytical solution may lie in the near injection region, where the relatively coarse discretization can lead to an increased numerical dispersion. However, with an outer domain radius of 100 km and a horizontal discretization length of $300\,\text{m} \times 300\,\text{m}$, the results are still in good agreement, with only a 3.3 % deviation from the Zeidouni-Method at the end of the injection, for both the Dirichlet and the Neumann scenario. The horizontal discretization length for the complex geological model is $300\,\text{m} \times 300\,\text{m}$ as previously discussed in Sec. 5.2. Given the results shown here, this horizontal discretization length is a good compromise between model accuracy, the need to sufficiently resolve the geology, and the computational

feasibility.